# DNA viral community enhances microbial carbon fixation capacity via auxiliary metabolic genes in contaminated soils

Jia-nan Lu[1], Yuanqing Chao [1] ✉, Li Tian[1], Xi Zhong[1], Ziwu Chen[1], Huan He[2], Bi Huang[3], Mengyao Li[1], Zekai Feng[1], Huayuan Feng[1], Chang Hu[1], Shunkang Zhou[2], Liqi Zhang[4], Yulu Yang[1], Zhepu Ruan [2], Kengbo Ding[1], Ying Yang[4], Ke Yuan[4], Wenshen Liu[1], Hua Qi[2], Yue Cao [1], Ying-heng Fei[5], Ning Ling[6], Shizhong Wang[1], Ye-tao Tang[1], Tiangang Luan[3], Zhihong Xu[7] & Rongliang Qiu [1,2] ✉

Soil is the largest organic matter repository on land and the virosphere is an essential component of soil carbon cycling. While a few carbon-related auxiliary metabolic genes (AMGs) in viruses are reported to potentially influence the hosts, the effects of virus-host interactions on soil carbon fixation, particularly in carbon-deficient contaminated soils, need further validation. Here, we explore the impact of viruses on carbon fixation in contaminated soils from 58 metal mining areas across eastern China. Eleven different functional categories of carbon fixation AMGs are identified via metagenomic analysis in 323 contaminated soil samples. Enzymatic activities of three key AMGs (i.e., *rbcL*, *ppdK* and *TKT*) are experimentally characterized, indicating the positive role of these genes in carbon fixation. Furthermore, transcriptomic sequencing reveals that after active virus inoculation the carbon fixation genes significantly up-regulate (~73%, $p < 0.05$). In mesocosms with stable isotope labeling, the accumulation of $^{13}$C-labeled organic carbon significantly increases (~10%, $p < 0.01$). Our results provide theoretical and experimental evidence for incorporating viral contributions into the assessments of carbon fixation, and improve the understanding of viral roles within the processes of carbon cycling.

Soils account for the largest pool of terrestrial carbon (C) (~1500–2400 Pg C)[1] and exert a crucial role in balancing global C cycle[2]. In spite of this, global soil C loss still remains a major concern. Many factors, such as global warming[3], land cover change[4,5], and biodiversity loss[6], may reduce soil C storage.

Soil contamination caused by mining activities may have a severe impact on the global C cycle through land cover change[7], which would be one of the main drivers of soil C loss[8]. Long-term soil contamination would lead to significant losses of soil organic C (SOC) stored in topsoil of mining areas[9] and further negatively affect the composition and diversity of microbial communities[10].

As the major agents of global SOC cycling, terrestrial soil microbes annually fix up to 4.1% of carbon dioxide ($CO_2$) from global atmosphere[11]. Then, soil microbial ecosystem severely damaged by

[1]School of Environmental Science and Engineering, Guangdong Provincial Key Laboratory of Environmental Pollution Control and Remediation Technology, Sun Yat-sen University, Guangzhou, China. [2]Guangdong Provincial Key Laboratory of Agricultural & Rural Pollution Abatement and Environmental Safety, College of Natural Resources and Environment, South China Agricultural University, Guangzhou, China. [3]State Key Laboratory of Biocontrol, School of Life Sciences, Sun Yat-sen University, Guangzhou, China. [4]School of Marine Sciences, Sun Yat-Sen University, Zhuhai, China. [5]School of Environmental Science and Engineering, Guangzhou University, Guangzhou, China. [6]State Key Laboratory of Herbage Improvement and Grassland Agro-Ecosystems, College of Pastoral Agriculture Science and Technology, Lanzhou University, Lanzhou, China. [7]School of Environment and Science—Chemistry and Forensic Science, Griffith University, Brisbane, QLD, Australia. ✉e-mail: chaoyuanq@mail.sysu.edu.cn; qiurl@scau.edu.cn

contamination would not be able to fulfill its role in the C fixation[12,13]. Contamination generally inhibited the Wood-Ljungdahl (WL) pathway[14,15] and the reduction of *Proteobacteria* abundance weakened the C fixation of reduced tricarboxylic acid (roTCA) cycle in long-term contaminated soils[16]. These imply that it is critical to enhance microbial C fixation and storage in contaminated soils. However, few studies establish the associations between microbes and viruses in contaminated soils, and the effects of microbial C fixation affected by viruses are also rarely involved.

Viruses are critical components in natural microbial communities, regulating population dynamics of prokaryotes[17] and shaping microbial evolution[18]. C accumulation caused by the cross-kingdom interactions between viruses and bacteria has recently garnered extensive attention[19]. During lytic infection cycles, overall virus-mediated recycling of organic matter may accelerate both respiration via the viral shunt and C storage through the microbial C pump[20,21]. With the environmental variations, different viral mechanisms to soil C cycling are likely to fluctuate[22,23]. In contaminated soils, more mutually beneficial virus-host relationships occurred with an increased probability of lysogenic infection, which further suggests that viruses may participate in the microbial C cycle through lysogeny[24].

Remarkably, lysogenic viruses may have further effects on the soil C cycle by reprogramming host metabolism by encoding C fixation auxiliary metabolic genes (C-fixation AMGs). For example, in deep sea sediments, viruses encoded AMGs such as transketolase (*TKT*) and pyruvate orthophosphate dikinase (*ppdK*) in Calvin Benson [CB] cycle and roTCA cycle, respectively[25,26]. Viruses in great salt lake contained ribulose-bisphosphate carboxylase large chain (*rbcL*) involved in CB cycle for C fixation[27]. In soil ecosystems, virus-mediated potential C-fixation AMGs, including *TKT*, ribose 5-phosphate isomerase A (*RpiA*), and ribose-phosphate pyrophosphokinase (*PrsA*), were also reported[28]. The above reports implied viruses could facilitate C fixation of their prokaryotic hosts through the AMGs expression. Despite these insights, the specific roles of viruses with different life cycles in

microbial-driven C fixation, especially their potential to enhance C fixation in contaminated soils, remain largely unexplored.

In this work, we estimate the positive effects of viruses on C fixation in contaminated soils through spatial investigations at the continental scale combined with experimental verification. Viral and prokaryotic genomes are extensively recovered from contaminated soils in 58 metal mining areas across eastern China. Eleven different types of C-fixation AMGs are identified in the contaminated soils and their functional activities are verified by protein expression. The significantly positive effects of viral inoculation on the microbial C-fixation capacity and the accumulation of ¹³C-labeled organic carbon are quantified by stable isotopic labeling microcosm incubation. Overall, our study provides critical insights on the positive effects of viruses on C fixation in the contaminated soils, and reinforces the necessity to incorporate viral contributions into assessments of C fixation across ecosystems.

## Results

### Abiotic and biological associations of viruses in contaminated soils

Based on large-scale spatial investigation, we collected a large number of soil samples around mining areas in eastern China, including 323 contaminated and 86 non-contaminated samples (Supplementary Data 1). In total, 31,210 viral operational taxonomic units (vOTUs, ≥5 kb) were collected from the contaminated soils and 1832 vOTUs were collected from non-contaminated soils using VirSorter (v1.0.5) and VIBRANT (v1.2.1) (Supplementary Figs. 1–3 and Supplementary Data 2–5). In the contaminated soils, nearly 70% of the viruses remained unclassified at the family level (Fig. 1b). Similar to non-contaminated soils, most classified viruses were taxonomically assigned into three families (*Mesyanzhinovviridae, Peduoviridae,* and *Casjensviridae*) (Fig. 1b) in contaminated soils. There were 4549 viral clusters (VCs) generated from the cluster of contaminated soils collected from IMG/VR v4 database and metallic mine soils (Fig. 1d and Supplementary Data 6 and 7). All selected ecosystems shared no more

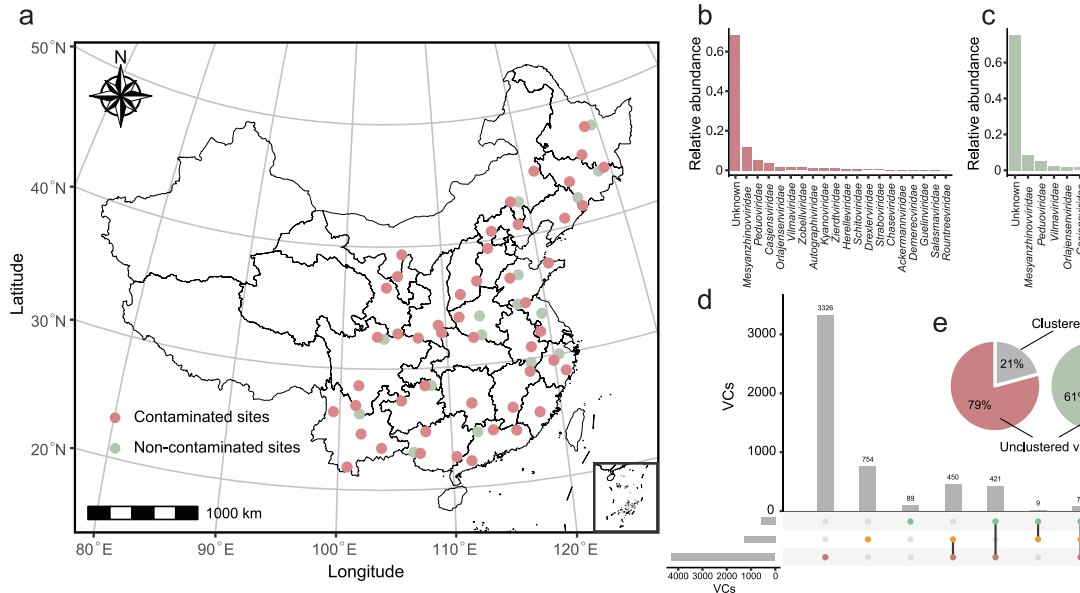

**Fig. 1 | Overview of viruses in metal mining areas. a** Geographical illustration of soil sampling sites in this study on the map of China. Detailed information of each sampling site is provided in Supplementary Data 1. Contaminated sites (*n* = 48) are marked by red dots and non-contaminated sites (*n* = 17) are marked by green dots. The map of China was obtained from Standard Map Service of the Ministry of Natural Resources, People's Republic of China [Map Review Number: GS(2024) 1158] and was visualized by the R package sf. Histogram showing the relative

abundance of viral species at family level in contaminated (**b**) and non-contaminated (**c**) sites. **d** Shared vOTUs with public databases. The UpSet diagram shows the sharing of mining viruses with the vOTUs of the public databases. These vOTUs in IMG/VR- contaminated soils were screened from the IMG/VR v4 database. **e** The proportion of these vOTU clusters is in the contaminated and non-contaminated sites.

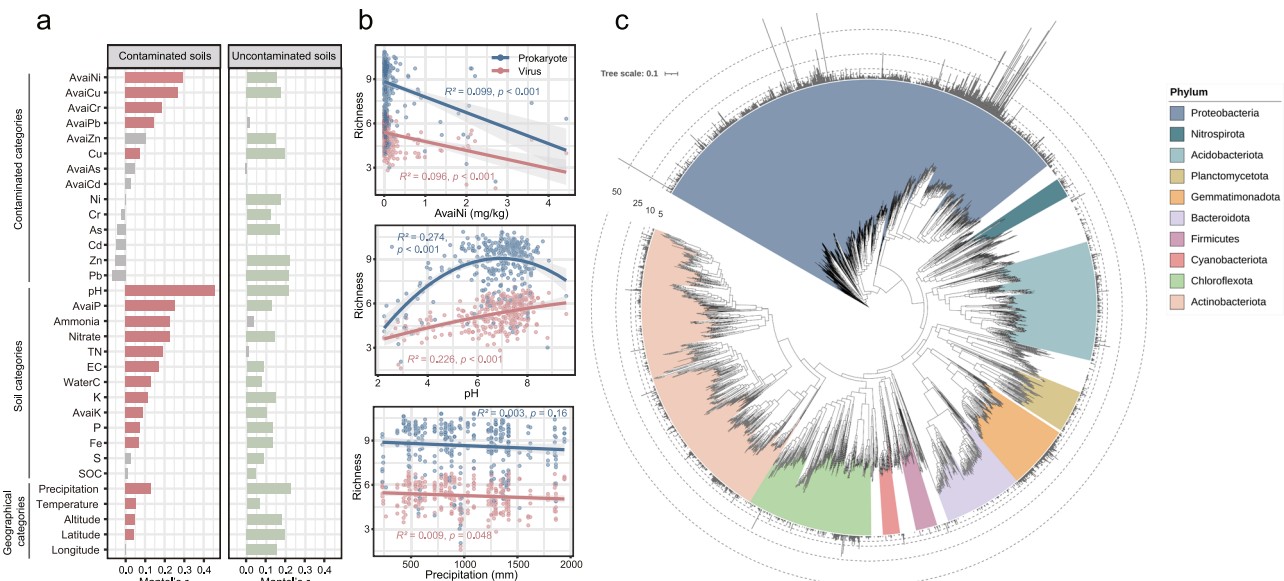

**Fig. 2 | Environmental variations of viral populations and virus-host linkage.**
**a** Mantel test of abiotic variables and viral communities. Variables that are significant are marked in red (contaminated sites) or green (non-contaminated sites). EC electronic conductivity, SOC total organic carbon, TN total nitrogen, WaterC water content, AvaiP available phosphorus, AvaiK available potassium.
**b** Relationships between richness of prokaryote and virus with abiotic variables.

The error bands represent the estimated 95% confidence interval. Statistical significance for regression coefficients was assessed using a two-sided $t$-test ($n = 323$). **c** Phylogenetic tree and viral population distribution based on high quality MAGs. Taxa of ten major phylum-level bacterial with high abundance are annotated. The outer bar chart represents the number of vOTUs for each MAG.

than 65 VCs in total. In the contaminated soils, 5001 vOTUs were clustered into 942 VCs with other ecosystems and 26,209 vOTUs were clustered alone (3326 VCs) or unclustered (Fig. 1c, e). The proportion of unique vOTUs in the contaminated soils was as high as 79%, while that in the non-contaminated soils was lower as 61% (Fig. 1e). Viruses in the contaminated soils were highly heterogeneous with the viruses in the non-contaminated soils or in the database. This indicated that the viral community in metallic mine soils may have endemic and unexplored diversity.

There were 20 factors which were significantly correlated with the structure of viral community in contaminated soils, including geographical (e.g., precipitation, temperature and altitude), soil physicochemical (e.g., pH, available phosphorus, and ammonia nitrogen) and contamination parameters (e.g., available Ni, Cu and Cr). In terms of the viral community, the effect of soil physicochemical parameters and heavy metal contamination indices was significantly greater in contaminated soils than it was in non-contaminated soils. (Fig. 2a). The pH (Mantel'$r$ = 0.46, $P < 0.001$) showed the strongest correlation with viruses, followed by available concentration of nickel (Mantel'$r$ = 0.30, $P < 0.001$) and available concentration of copper (Mantel'$r$ = 0.27, $P < 0.001$) (Fig. 2a). Most of the available concentration of heavy metals exhibited stronger effects on the viral community than the total concentration of heavy metals in the contaminated soils (Fig. 2a).

The environmental factors of the highest correlation with viral community in contaminated soils were selected as representatives for regression analysis with prokaryotic and viral richness. The response of viral richness to precipitation was more sensitive than that of prokaryotes in the contaminated soils (Fig. 2b). The pH was strongly correlated with prokaryotic and viral richness, and the variation of virus diversity with pH is smaller than that of prokaryotes (Fig. 2b). With the increase of contamination (e.g., available Ni), the richness of prokaryotes and viruses showed a decreasing trend significantly (Fig. 2b).

To further investigate potential associations between viruses and prokaryotic hosts, 6904 prokaryotic metagenome-assembled genomes (MAGs, ≥ 70% genome completeness and <10% contamination) were recovered from the contaminated soil metagenomes

(Supplementary Data 8 and 9). The results showed that 4781 viral genomes generated 19,293 virus-host associations with 4489 prokaryotic MAGs in the contaminated soils (Supplementary Data 10 and 11). In the contaminated soils, the hosts of viruses were widely distributed across 36 prokaryotic phyla, mainly including *Proteobacteria* ($n = 1740$), *Actinobacteriota* ($n = 919$), and *Acidobacteriota* ($n = 474$) (Fig. 2c). Observing that virus-host associations are not evenly distributed across the phylogenetic tree (Fig. 2c), we calculated the viral range (defined as the number of viral populations that are associated with a single host population) (Supplementary Figs. 4 and 5). Interestingly, *Patescibacteria* with low relative abundance (0.75%) had the highest viral range of 8.39 (Supplementary Fig. 4b). Similar results could be found in *Dependentiae* (viral range = 5, relative abundance = 0.01%) and *Crenarchaeota* (viral range = 4.13, relative abundance = 3.18%), indicating that viral infection had a typical host preference in the contaminated soils (Supplementary Fig. 4b).

## Virus-encoded AMGs involved in C fixation

In general, viruses tend to harbor AMGs for amino acid (1903 AMGs), cofactor/vitamin (1616 AMGs) and carbohydrate (1513 AMGs) metabolisms in the contaminated soils (Supplementary Fig. 6a and Supplementary Data 12). In contaminated soils, 23 AMGs belonging to 11 types were identified to be involved in C fixation (Supplementary Data 13). The predicted hosts of viruses carrying these AMGs were distributed in 4 bacterial phyla including *Proteobacteria*, *Planctomycetota*, *Acidobacteria* and UBP10 (Supplementary Fig. 6b). Noticeably, these AMGs covered several key C-fixation pathways, including the CB cycle, roTCA cycle, 3-hydroxypropionate/4-Hydroxybutyrate cycle (3-HP/4-HB cycle), DC/4-HB cycle, 3-hydroxypropionate bicycle (3-HP bi-cycle), and WL pathway (Fig. 3a). Remarkably, *rbcL* gene (ribulose-bisphosphate carboxylase large chain) and *korB* gene (2-oxoglutarate/2-oxoacid ferredoxin oxidoreductase subunit beta) are key determinants of C assimilation rate in the CB cycle and roTCA cycle, respectively. The abundance of viruses carrying them accounted for 18% and 15% of the abundance of viruses with C-fixation AMGs in the contaminated soils (Fig. 3a). Additionally, based on aggregated boosted tree analysis (Fig. 3b), the viruses carrying C-fixation AMGs were found

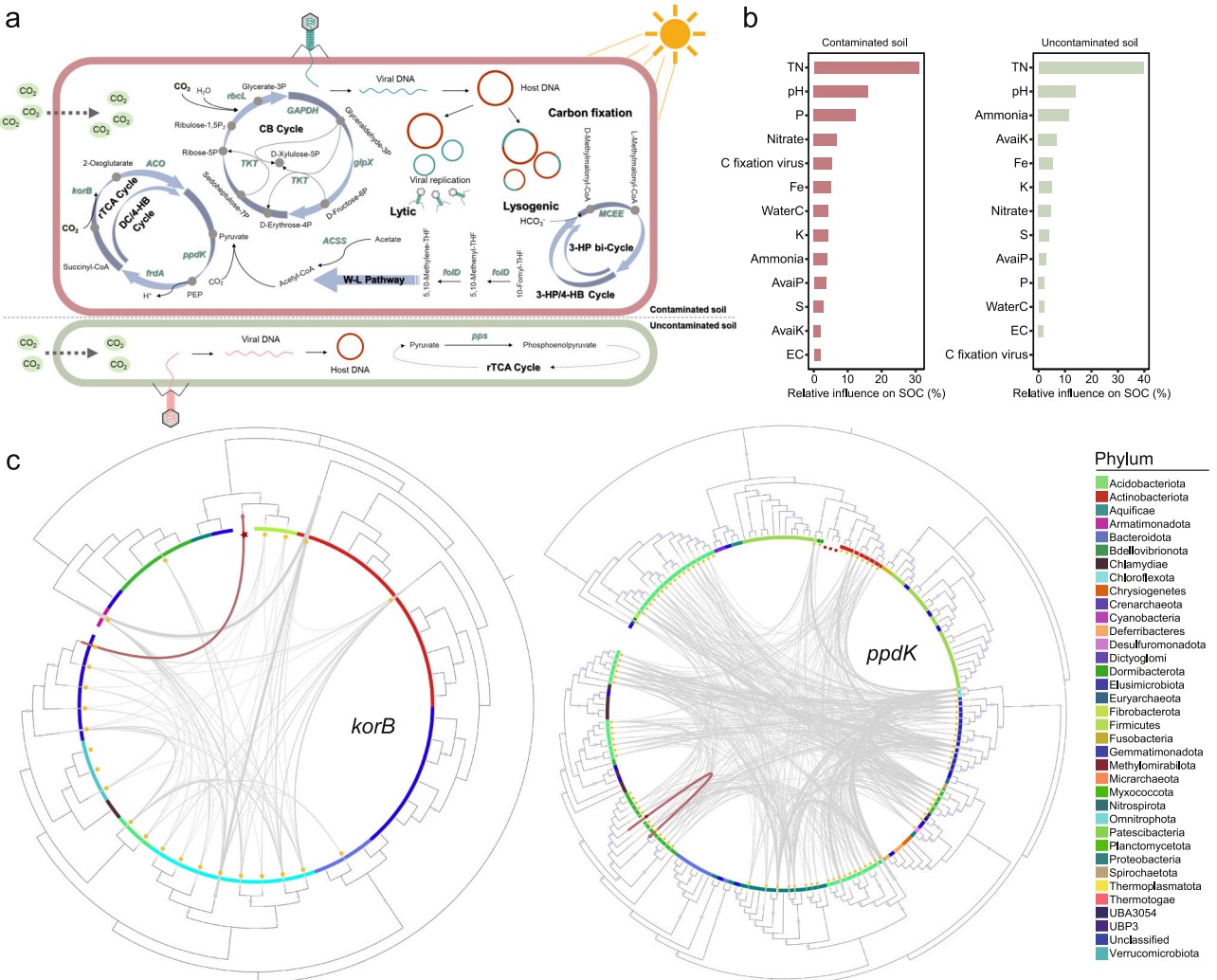

**Fig. 3 | Function and environmental distribution of carbon fixation AMGs.**
**a** Several types of C-fixation AMGs carried by the virus. The comparison of C fixation types in polluted and non-polluted habitats is shown through pathways. Six types of C fixation pathways were involved in contaminated soils. **b** Aggregated boosted tree (ABT) models was used to evaluate the relative influence of selected factors on SOC. EC electronic conductivity, TN total nitrogen, WaterC water content, C fixation virus, relative abundance of virus carrying C-fixation AMGs in the metagenome. **c** Phylogenetic tree of AMGs. Different lineages of AMGs from bacteria are assigned different range colors according to their taxonomic relationships, as shown in the legend. The red stars and yellow dots represent sequences obtained from mining viruses and MAGs in this study, respectively. The gray lines indicate the predicted horizontal gene transfer events that occurred, and the red lines indicate the exact horizontal gene transfer that occurred between the viral AMGs and MAGs in situ.

to have considerable influence (5%) on SOC in contaminated soils. By contrast, the influence of viruses carrying C-fixation AMGs on SOC was not detected in the non-contaminated soils (Fig. 3b).

In addition, the above two key genes (*rbcL* and *korB*) in the C-fixation pathways (Fig. 3a) and the two most abundant AMGs (*ppdK* in roTCA cycle and *TKT* in CB cycle) were selected to further explore the origin of these AMGs (Supplementary Data 13). Homologous genes with the selected AMGs from eggNOG database (v5.0.0) and prokaryotic MAGs were obtained and merged to construct phylogenetic trees (Fig. 3c and Supplementary Fig. 7). These results revealed that *rbcL* genes were clustered primarily with homologs from *Proteobacteria*. Similarly, *ppdK* genes were predominantly affiliated with homologs originating from *Planctomycetota*. This finding aligned with our previous host predictions (Supplementary Fig. 6b), indicating that these AMGs may be closely tied to host distribution of the viruses. Notably, the phylogeny of these homologs did not correlate well with the taxonomic assignment of microbial species (Fig. 3c), suggesting a significant role of horizontal gene transfer (HGT) in shaping their distribution. Consequently, we quantified HGT events among MAGs harboring these homologs and viral genomes encoding AMGs. Overall,

there were numerous HGTs between MAGs both within one phylum and across phyla. While, specific HGTs of AMGs exactly occurred between MAGs and viruses, e.g., *korB* transferred from *Elusimicrobiota* to *Mesyanzhinovviridae* and *ppdK* transferred from *Vilmaviridae* to *Planctomycetota* (Fig. 3c).

Most of the C-fixation AMGs were located between two viral marker genes, including *rbcL1*, *ppdK1*, and *TKT1* (Fig. 4a). Our analysis of the viral genomes revealed the presence of several prophage genes (e.g., phage major capsid protein and phage terminase large subunit) and phage genes with lysogenic potential (e.g., terminase large subunit and phage integrase family protein) (Fig. 4b). Among the 23 viral genomes carrying the C-fixation AMGs, 6 vOTUs were identified as prophages, while 7 vOTUs exhibited lysogenic potential (Fig. 4b). This suggested that AMGs could be integrated into the host genomes for C-fixation in cases where viruses coexist with the hosts. Based on this, we further predicted the structures of these AMGs by phyre2 (Fig. 4c). Although these AMGs shared low similarity in amino acid sequence (25–36%) with previously identified microbial genes, they exhibited remarkably similar tertiary structures (confidence = 100%) (Supplementary Fig. 8). A promoter (*p* = 0.0001) was identified at the

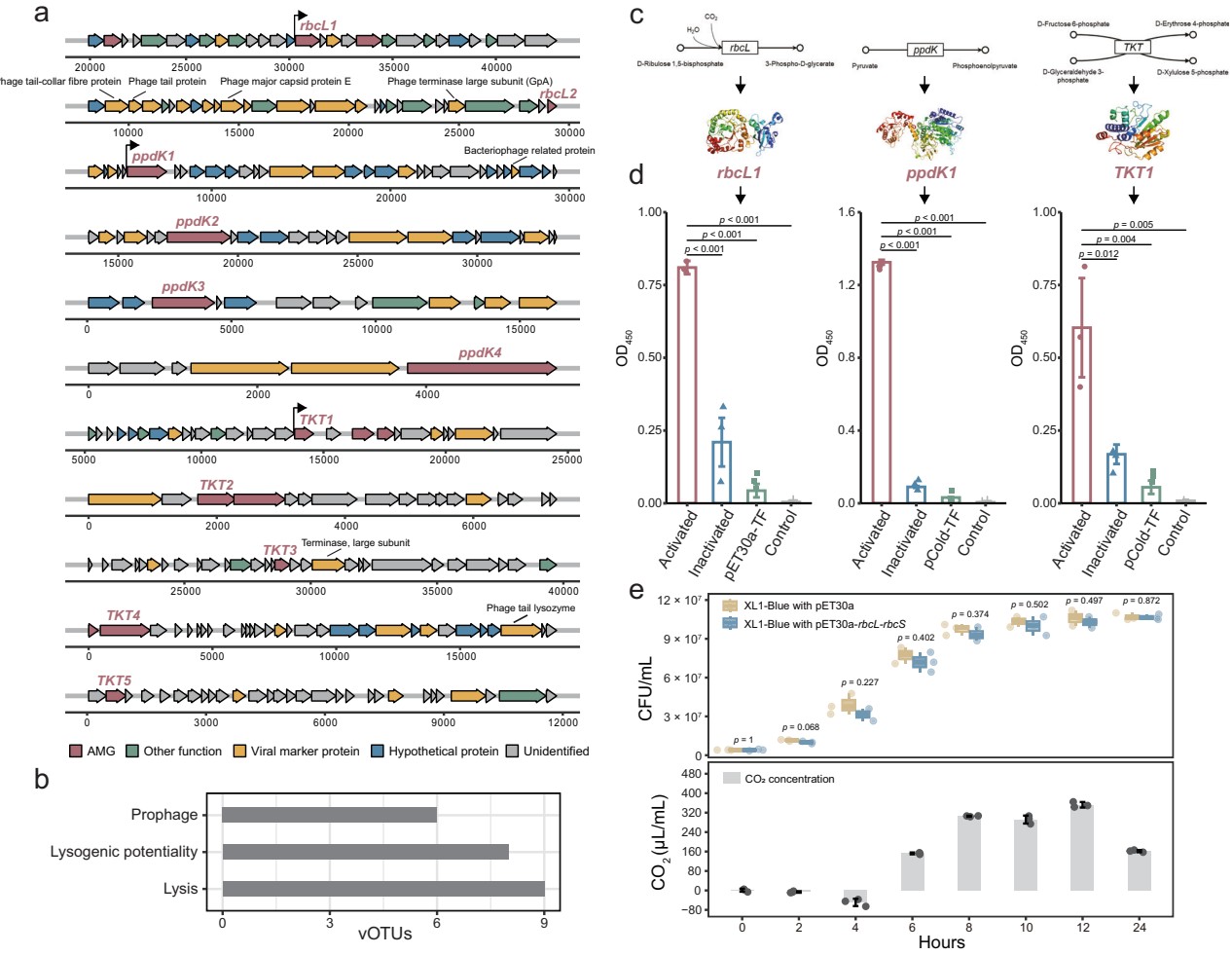

**Fig. 4 | Characterization of auxiliary metabolic genes (AMGs) in contaminated soil viruses. a** Genome map of viruses carrying C-fixation AMGs. Arrows depict the position and orientation of predicted proteins on the viral genome, and fill colors represent different functional classes of genes, as shown in the legend. AMGs, which encoded by viruses, are shown in red. The arrows represent the promoter. **b** Bar chart of lifestyle distribution of viruses with C-fixation AMGs. **c** Chemical reaction process mediated by C-fixation AMGs. **d** Enzyme activity determination of C-fixation AMGs. Data are presented as mean values ± standard deviation (SD).

Statistical significance for data analysis was assessed using a two-sided $t$-test ($n = 3$). **e** Growth curves of bacteria and the temporal variations of the $CO_2$. The $CO_2$ concentration is represented by the difference generated by XL1-Blue with pET30a-$rbcL$-$rbcS$ minus XL1-Blue with pET30a. Boxes show medians/quartiles. Whiskers extend to the most extreme vales within 1.5 interquartile ranges. Points shown beyond the whiskers are outliers. Data are presented as mean values ± SD. Statistical significance for data analysis was assessed using a two-sided $t$-test ($n = 3$).

upstream ends of $rbcL1$, as was observed in $ppdK1$ ($p = 0.0001$) and $TKT1$ ($p = 0.001$) (Fig. 4a).

To validate whether the AMGs exhibit functional activity, gene synthesis and expression were conducted to characterize their enzymatic activities in laboratory with three controls including blank, pCold-TF/PET-30a protein-free, and high-temperature inactivation (Fig. 4d and Supplementary Fig. 9). These AMGs all showed high enzyme activities (Fig. 4d). Co-expressed with $rbcS$, $rbcL1$ was active in cell lysate, which catalyzed the primary carboxylation of ribulose 1, 5-diphosphate and fixed $CO_2$ to the biomass. In addition, using VCSM13 helper phage as a vector, C-fixation AMGs (i.e., $ppdK1$, $TKT1$ and $rbcL1$) from *E. coli* TG1 were successfully transduced into *E. coli* XL1 Blue. Gene transfer was confirmed by antibiotic screening and PCR amplification (Supplementary Fig. 10). This result highlighted the crucial role of viruses as genetic vector in promoting horizontal transfer of C-fixation genes. Given the active co-expression of $rbcL1$ and $rbcS$, *E. coli* XL1 Blue emitted less $CO_2$ (Fig. 4e), indicating that the heterotrophic *E. coli* XL1 Blue obtained C-fixation ability by transferring relevant AMGs. Together, these findings suggested that the identified AMGs were of viral origin and may integrate into host genomes for transcription with a potential contribution to host C fixation during infection.

## Impact of viral inoculation on C fixation in soils

The above bioinformatics and molecular biology results indicated that viruses may play a positive role in C fixation of the contaminated soils. For further verification, a viral community inoculation experiment with $^{13}C$ labeled inorganic C source was conducted to explore the effects of viruses on contaminated soil C fixation. After active virus inoculation, soil inorganic C (SIC) demonstrated a consistent decline (Fig. 5a). Compared to the inactive inoculation, the treatment inoculated with active viruses exhibited a more substantial reduction of SIC (-10%, $p < 0.01$, Fig. 5a). Meanwhile, SOC significantly increased by -11% ($p < 0.05$), representing a 5.8-fold higher increase than the inactive virus treatment (-1.9%) (Fig. 5b). Notably, active virus inoculation exhibited greater $^{13}C$-SOC by -10% ($p < 0.01$) compared to inactive virus treatment (Fig. 5d), suggesting the enhanced accumulation of SOC was from the labeled inorganic C source. These results provided direct evidences that viruses could significantly promote C fixation. Additionally, it was observed that the expression of $rbcL1$, $ppdK1$, and $TKT1$ significantly increased after active virus inoculation (Fig. 5e–g). The abundance of C-fixation genes in the active virus treatment was also significantly higher (-73%, $p < 0.05$) than that in the inactivated virus treatment (Fig. 5h). Combining the results of soil C turnover and genes

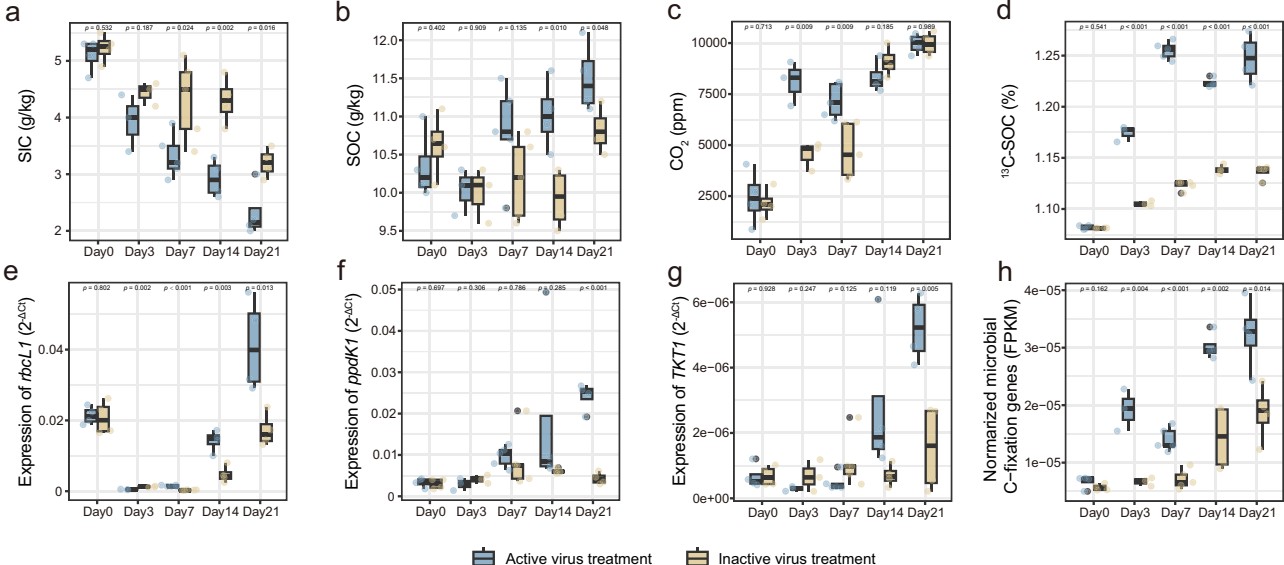

**Fig. 5 | Temporal dynamics of soil carbon turnover, AMGs, and microbial functional genes after active and inactive virus inoculation.** Changes of SOC (**a**), SIC (**b**), $CO_2$ (**c**) and $^{13}$C-SOC (**d**) after viral inoculation. Variation of AMGs, including *rbcL1* (**e**), *ppdK1* (**f**) and *TKT1* (**g**), and microbial C-fixation genes (**h**) after viral inoculation. Microbial C-fixation genes in each metabolic pathway were first averaged to derive the within-pathway mean (e.g., average FPKM for Calvin cycle genes), followed by a secondary averaging of pathway-specific means to mitigate gene count bias. Boxes show medians/quartiles. Whiskers extend to the most extreme values within 1.5 interquartile ranges. Points shown beyond the whiskers are outliers. Statistical significance for data analysis was assessed using a two-sided *t*-test (*n* = 5).

expression, the role of viral AMGs in strengthening C fixation was highlighted.

Moreover, we investigated the effect of viral inoculation on the C fixation capacity of microorganisms. The results showed that there was no significant change in the abundance of C-fixation genes on the first day after viral inoculation (Fig. 6a). After 1 week of inoculation, the ratio of up-regulation to down-regulation of C-fixation genes was increased compared with that of the inactive viruses (Fig. 6a). These results suggested that viral inoculation significantly increased the C-fixation capacity of microbial community.

Corresponding to the AMGs identified in previously contaminated soils, abundance of genes in the C-fixation pathways also changed with viral inoculation. In the CB cycle, *GPDH*, *TKT*, and *glpX* all showed significant up-regulation on day 3, and *GPDH* showed significant enrichment in subsequent observations (Fig. 6b). Although *rbcL* was down-regulated on day 3, it also began to become enriched on day 7 (Fig. 6b). It was worth noting that *accA/B/C* were significantly enriched after viral inoculation, which were related to C fixation in the 3-HP bi-cycle and 3-HP/4-HB cycle. Significant enrichment of *por* associated with roTCA cycling also occurred at day 14 after inoculation.

Soil microbial communities also showed significant differences after viral inoculation (Supplementary Data 14). The abundance of some autotrophic bacteria increased, such as *Xanthobacteraceae*, *Rhodanobacteraceae* and *Nitrospiraceae* (Fig. 6c). Compared with the inactive inoculation, the relative abundance of *Xanthobacteraceae* and *Rhodanobacteraceae* in the treatment of active inoculation increased from 0.039% and 0.023% to 1.01% and 3.23%, respectively. Consistent with previous studies on metagenomics, *Xanthobacteraceae* had a high proportion (21%) in hosts of viruses carrying C-fixation AMGs (Supplementary Fig. 6b). *Rhodanobacteraceae* in *Proteobacteria* with the highest abundance and *Nitrospiraceae* in *Nitrospirota* in the contaminated soils, which had a high viral infection rate (viral range = 2.38) (Fig. 2c and Supplementary Fig. 4b). In short, this experimental verification showed that viral inoculation significantly promoted the expression of C-fixation genes in the microbial communities and accelerated SOC accumulation in the contaminated soils.

## Discussion

The viruses in the contaminated soils exhibited immense and unexplored species diversity with a higher richness than soil bacteria (Supplementary Fig. 4a). Over 60% of vOTUs remained unannotated at family level (Fig. 1b, c), underscoring the unique composition of viruses in the contaminated soils. About 84% of viruses in the contaminated soils remain highly unknown (Fig. 1d), which is consistent with previous studies on inorganic[29] and organic[30] contaminated soils. Additionally, the low dispersal capability of soil viruses led to a decrease in viral community similarity with increasing spatial distance (Supplementary Fig. 11). Few vOTUs and VCs were shared among different ecosystems (Fig. 1e), further revealing a high spatial turnover rate and within viral communities[28,31] and uniqueness of the viral ecological niche[32]. Hindered by the vast "dark matter" of viruses and microorganisms, further exploration is needed to clarify the classification and functional attribution of soil viruses[33,34].

Viral infection is ubiquitous in various ecosystems (e.g., terrestrial and aquatic ecosystems)[17,18,35], wherein viral expression of AMGs diverts host metabolisms and could enhance the adaptability of both hosts and viruses in the soil[36]. Despite this, the relationship between viral auxiliary metabolism and microbial C fixation from the environment has not been fully explored. To our knowledge, 3 types of C-fixation AMGs (i.e., *ppdK*, *TKT* and *rbcL*) have been discovered in aquatic environments such as deep seas[25,26] and salt lakes[27], involving the CB cycle and roTCA cycle. In terrestrial ecosystems, a global atlas of soil viruses revealed that AMGs (i.e., *TKT*, *RpiA* and *PrsA*) related to C fixation might participate in the CB cycle[28], but there is a lack of direct evidence to demonstrate the role of AMGs in C fixation. In the present study, through continent-scale metagenomic investigations, we totally identified 12 types of C-fixation AMGs with 11 of them present in virus genomes in the contaminated soils (Fig. 3a). Among them, 8 types (i.e., *korB*, *frdA*, *ACO*, *GAPDH*, *glpX*, *ACSS*, *folD*, and *MCEE*) were not previously reported in viral AMGs, greatly expanding the diversity of C-fixation AMGs. However, several AMGs in the roTCA cycle, including *korB*, *frdA* and *ppdK*, could occur in the TCA cycle, as the roTCA cycle shares all enzymes with the TCA cycle[37,38]. The AMGs *korB* and *frdA* were also involved in glutamate biosynthesis[39,40] and fumarate

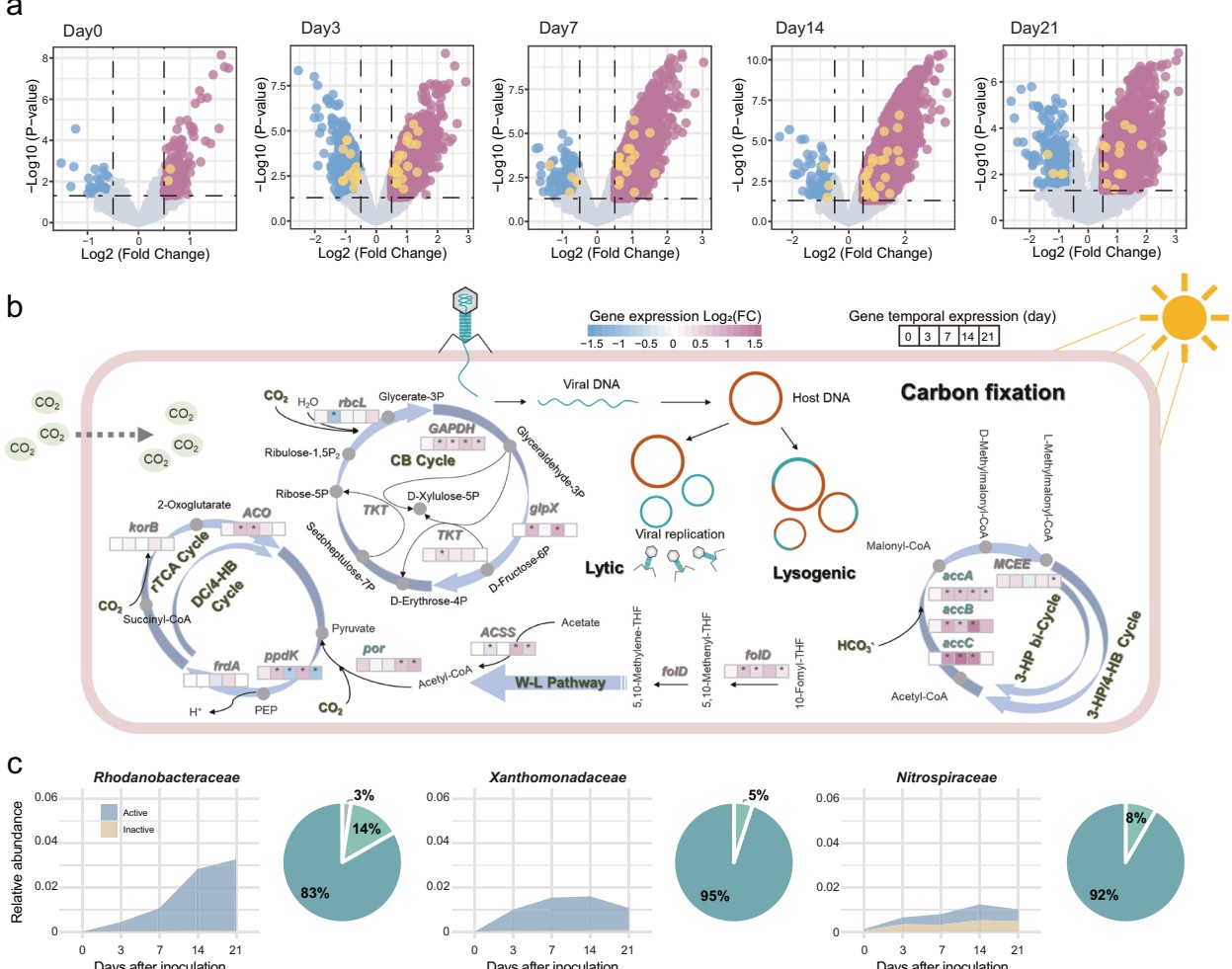

**Fig. 6 | Transcriptional profile and microbial population changes with viral inoculation in contaminated soils. a** Expression of genes in samples inoculated with active viruses compared to controls. Statistical significance for data analysis was assessed using a two-sided $t$-test ($n = 5$). Gene abundance is significantly up-regulated or down-regulated with a log2-fold (fc) variation. When log2 (fc) >0, the gene in the sample is significantly upregulated compared to the control, and vice versa. Each circle represents a gene, purple for up-regulated, blue for down-regulated, and yellow for C fixation functional genes. **b** Temporal changes in gene expression of C fixation genes. **c** Variation in the relative abundance of the major groups of autotrophic bacteria and C fixation capacity of MAGs. The area map shows the change in the relative abundance of autotrophic bacteria after viral inoculation. In the pie chart, dark green shows that MAGs with $CO_2$ fixed genes, light green shows that MAGs with genes except $CO_2$ fixed genes in the carbon fixation pathway, and gray shows that MAGs with no C-fixation related genes.

respiration[41], respectively. Therefore, it should be noted that these viral AMGs may contribute to C fixation and also other metabolic pathways.

Intriguingly, we discovered that these C-fixation AMGs mostly originated from stressed environments. Under high pressure (e.g., deep sea[25]), high salinity (e.g., salt lakes[27]), drought (e.g., desert[42]) and heavy metal contaminations, more AMGs are found to be associated with C fixation, potentially facilitating the modulation of C loss under resource-limited conditions[43]. Meanwhile, serving as important pools of genes associated with microbial C fixation, viral communities have the capacity to microbial ecology and biogeochemistry in stressed environments[44]. Such genetic exchanges also contribute to the functional redundancy and resilience of microbial C fixation (Fig. 4d, e), as AMGs could compensate for genetic losses or environmental fluctuations. This would further provide their hosts with access to much-needed C sources and an adaptive advantage in acquiring essential nutrients[45,46].

Over an extended period of association, the fate of viruses may become intertwined with that of hosts[47,48]. Considering the host association of AMGs, various mutually beneficial virus-host

interactions are noted[18,49]. In the present study, autotrophic bacteria (*Xanthobacteraceae*, *Rhodanobacteraceae* and *Nitrospiraceae*) in the soil ecosystem was found to be positively affected by viruses (Fig. 6c). It was documented that chemoautotrophic bacteria are abundant and widely distribute in the contaminated soils under low pH and elevated toxic metal levels[50], and play a significantly positive role in soil C fixation[51]. Viruses can regulate and even promote the performance of soil chemoautotrophic communities, which will increase the accumulation of SOC and further enhance the overall resilient and adaptive capacities of microbiome in stressful environments[52].

Viral infections in chemoautotrophic hosts appear to enhance their productivity[35] and the maintenance of autotrophic ability also facilitates the replication of viruses[53]. This is likely related to the "making winners" strategy[48], when the virus inserts events through regulatory regions to control host gene expression, and beneficial "active lysogenicity" may help the host become a winner. In our results, at least 61% of the viruses carrying C-fixation AMGs exhibited lysogenic abilities (26% were lysogenic phages, and 35% had lysogenic potential) (Fig. 4b). Increasing recognition of the prevalence and potential fitness advantages of lysogeny suggest that non-fatal infection strategies

carry distinct consequences for C cycling[54-56]. When viral infection imparts new cellular-level characteristics, e.g., autotrophic C-fixation capabilities (Fig. 6a, b), that allow the viruses and hosts to proliferate in additional environments[57,58], virus-host interactions may evolve into mutualistic relationships[24,48]. Such synergy heightens the adaptability of microbial communities, ultimately leading to a more stable and efficient C fixation structure in soils.

Given the great importance of microorganisms in controlling terrestrial C flux[1,59], soil C cycle influenced by viruses closely associated with hosts should be considered, including C fixation, transformation and mineralization. At present, a series of studies focused on the impacts of viruses on reducing soil C mineralization[60] and accumulating the recalcitrant dissolved organic matter components[20]. Studies on the role of viruses in soil C fixation are rarely reported. In the current study, the increase of $^{13}$C-SOC provided direct evidence for virus-mediated C fixation (Fig. 5d). In contrast to "Viral Shunt"[61], the nature of the phenotypic modifications by viruses, such as increased autotrophs abundance (Fig. 6c) and up-regulated microbial C-fixation genes (Fig. 6a), may further directly affect C storage (Fig. 5b)[62]. In agricultural soils, glycosyl transferases, viruses involved in, also convert the bioavailable carbohydrates to more stable carbohydrates, promoting the C sequestration[63]. Overall, the new evidence for the important role of viruses in soil C accumulation by enhancing C fixation and the new ideas about evaluating the viral contribution on C fixation were provided.

Under the combined stress of the C deficiency and soil contamination, virus-mediated acquisition of nutrients and living space becomes paramount for the selection of microbial functions[30,64]. Viral inoculation reduced the C mineralization in contaminated soils and had positive effects on the maintenance of organic C pool[63]. Then, re-carbonization of the depleted SOC pool, which plays a fundamental role in supporting various functions, requires consistent input of biomass-C and essential elements[65]. The enhanced C-fixation capacity due to viral inoculation (Figs. 5b, d and 6a) would further promote overall C metabolism in contaminated soils. Similar to recent studies, biodegradable microplastics enriched high auxiliary C metabolism, which promoted microplastic-derived C into microbial biomass or related products[66]. In C deficient soils, viruses and AMGs could increase the soil C accumulation by driving organic C metabolic adaptations[64]. These findings collectively highlight the critical roles of viruses in adaptively modulating C cycle in response to environmental changes and potentially maintaining C reservoirs in stressed ecosystems. Our observations (Supplementary Fig. 12) were also supported by the previous model prediction that the microbial partitioning of C used in metabolism would have a positive correlation with SOC[67]. This further implies that microbial C metabolism promoted by viruses is more conducive to the accumulation of SOC. Taken together, our findings systematically emphasize the importance of soil virome in mediating soil C cycle and might promote the restoration of the contaminated soil ecosystem[68].

In summary, we reveal the diversity and host-specificity of virus-assisted metabolism related to C fixation, and further verify the enzymatic activities of three key C-fixation AMGs encoded by soil viruses. Viral inoculation stimulates the accumulation of SOC in contaminated soils, providing metatranscriptomic evidence for the important ecological role of viruses in influencing microbial C fixation. We systematically explored the positive impact of viruses on C fixation in contaminated soils through multiomics and large-scale spatial datasets integrated with validation experiments. These findings provide theoretical basis for enhancing soil C sink by viruses and accelerating ecological restoration of degraded habitat. Understanding the virus-host-environment interactions in extreme environments will contribute to the development of more effective management measures to solve the problem of C loss in the contaminated soils.

## Methods

### Sampling collection
From June to September 2017, an investigation at the continental scale was conducted in metal mining areas of eastern China (latitude 21 °N to 47 °N, longitude 99 °E to 129 °E, Fig. 1a). A total of 50 mining areas, comprising 16 iron mines, 18 copper mines and 16 lead-zinc mines, were selected in line with the National Mineral Resources Development Plan (2016–2020), as these three types of mines receive the largest amount of provincial exploration investment and rank highest in terms of ore production and consumption in China. At each selected site, 3–6 soil samples were collected from abandoned lands, tailing areas, downstream contaminated farmlands and non-contaminated sites, making a total of 409 samples. For each composite soil sample, five soil cores (collected from the 0-10 cm topsoil layer) were mixed. All samples were capsulated in zipped bags on-site and disposed promptly after being transported back to the laboratory. Specifically, samples for microbial experiments are collected in sterile tubes and stored at −80 °C upon return to the laboratory. Data on mean annual temperature, mean annual precipitation, and mean annual humidity were derived from daily records of the nearest weather stations affiliated with the National Meteorological Information Center (http://data.cma.cn/en, Supplementary Data 1).

### Soil physical and chemical analyses
Discrete plant residues and stones were first removed from all soil samples via hand-picking. After being air-dried, the samples were sieved through plastic meshes (830 μm and 150 μm) and stored at ambient temperature for subsequent physicochemical determinations. Soil pH was determined at a soil-to-water ratio of 1:2.5 (w/v) and electrical conductivity (EC) were examined by the $BaCl_2$ compulsive exchange method[69]. A SOC-TN analyzer (Vario SOC, Elementar, Germany) was used to measure soil organic carbon (SOC) and total nitrogen (TN). For the concentrations of $NH_4^+$-N and $NO_3^-$-N, soils were extracted with 1 M KCl and measurements were conducted in accordance with the standard protocols specified in National Environmental Protection Standards of China (HJ634−2012 and GB/T32737-2016). Assessment of soil moisture involved placing 5 grams of soil in a 105 °C oven, where samples were dried until their weight remained unchanged. The moisture content (%) was measured as the weight loss proportion (relative to the original weight) resulting from oven-drying. For total element (P, S, Pb, Zn, Cu, Cd, As, Cr, Ni) analysis, sub-samples of the processed soils were digested with aqua regia using a MARS6 microwave (CEM, USA), and the resulting digests were analyzed by ICP-OES (Optima 5300DV, Perkin-Elmer, USA)[70]. For evaluating available heavy metals, 2.0 g soil samples were extracted with 20 mL of 0.01 M $CaCl_2$ followed by a 3-h shaking period. The extract was centrifuged at $3000 \times g$ for 20 min at 25 °C, after which the supernatants were filtered through 0.45 μm membranes and transferred to 10 mL centrifuge tubes. Prior to analysis, they were acidified with $HNO_3$ and stored in a refrigerator at 4 °C[71]. Heavy metal concentrations in all extracts were determined via ICP-OES. The neutral salt $CaCl_2$ was selected as the extractant to mimic natural environmental conditions, as it effectively indicates the phytoavailability of various metals[71]. Standard reference soil samples, duplicate samples and reagent blanks were employed as quality control measures.

### DNA extraction and metagenomic sequencing
The extraction of DNA from 409 soil samples was carried out with the Fast DNA®SPIN kit (MP Biomedicals, France). Each extraction involved processing around 5 g of fresh soil with the Fast DNA®SPIN Kit for Soil (MP Biomedicals, France) in accordance with the manufacturer's protocol. The DNA concentration and purity were evaluated by NanoDrop 2000 (Thermo Fisher Scientific, USA). Ultimately, the purified DNA samples from all 409 soil samples were sequenced on the Illumina

NovaSeq platform (350 bp insert size, paired-end 150 bp) by Novogene (Beijing, China).

## Metagenome assembly and prokaryotic genome binning

Illumina raw sequences underwent preprocessing using Trimmomatic v0.35, with adaptor sequences and low-quality regions being removed in the process[72]. For each soil sample, quality-controlled reads were assembled using MEGAHIT v1.2.9[73] with parameters "-k-min 27 -k-max 127 -k-step 10 -min-contig-len 500". MAGs were generated by binning contigs from individual assemblies (parameters: --maxbin2 --metabat1 --metabat2) and Bin_refinement module (parameters: -c 70 -x 10)[74]. All medium-to-high quality MAGs (completeness > 70%, contamination < 10%) were dereplicated using dRep v2.6.2 (parameters: -comp 70 -con 10 -sa 0.95)[75]. Taxonomy of each MAG was identified by GTDB-Tk v0.2.2[76] (https://github.com/Ecogenomics/GtdbTk).

## Identification, taxonomic annotation and lifestyle prediction of viral contigs

Viral contigs were identified from metagenomic assemblies using Vir-Sorter v1.0.5[77] and VIBRANT v1.2.1[78], which from both VirSorter categories 1, 2, 4, and 5 and VIBRANT without fragments were retained. All identified contigs ≥5 kb were clustered into vOTUs using CD-HIT-EST v4.8.1[79] with "-c 0.95 -aS 0.9". CheckV v0.6.0[80] was employed to assess the viral genome integrity and to predict the life strategies (virulent or temperate) in combination with VIBRANT v1.2.1. The taxonomic affiliations of the vOTUs were identified by PhaGCN2[81]. Virus sequences (≥5 kb) for similar environments were collected from the IMG/VR4 database[82]. Open reading frames (ORFs) were predicted by Prodigal v2.6.3[83], and protein-sharing network analysis of viral populations from metallic mine soil and public database was performed by vConTACT2[84].

## Abundance profiles

The trimmed reads from each sample were mapped to viral contigs and MAGs with BamM v1.7.3 "make" (https://github.com/Ecogenomics/BamM). Low-quality alignments were removed using BamM v1.7.3 "filter" mode (parameters: --percentage_id 0.95 --percentage_aln 0.75). Filtered bam files were subsequently submitted to CoverM v0.3.1 (https://github.com/wwood/CoverM) to produce sample-specific coverage profiles using contig mode applied to viral contigs and genome mode applied to MAGs (parameters: --trim-min 0.10 --trim-max 0.90 --min-read-percentage-identity 0.95 --min-read-aligned-percent 0.75 -m tpm).

## Prediction of virus-host linkages

Three silico approaches were applied to predict virus-host interactions, including nucleotide sequence homology, transfer RNA (tRNA) matching and CRISPR-spacers matching. Briefly, all viral contigs were aligned with the prokaryotic MAGs using BLASTn (bitscore ≥50, $e \le 10^{-3}$, identity ≥70%, and matching length ≥2500 bp)[85]. The tRNAs from viral contigs and prokaryotic MAGs was identified using ARA-GORN v1.265 with the "-t" option[86], after which perfect virus-host matches were selected. CRISPR spacers were extracted from prokaryotic MAGs using CRISPRCasTyper with default parameters[87], then aligned to the extract of viral contigs via BLASTn under the thresholds of $E \le 10^{-5}$ with 100% perfect matches over the complete spacer length.

## Identification, phylogenetic analysis and HGT prediction of viral AMGs

ORFs of viral contigs were preliminary annotated based on the egg-NOG v5.0.0 database[88]. In a further step, annotations of viral contigs were also conducted using VIBRANT and DRAM-v[89] with KEGG[90], Pfam[91] and VOG[92] database as references and default parameters applied. The "metabolic pathways" and "sulfur relay system" of KEGG annotations categories were reported as potential AMGs.

Furthermore, these AMGs were annotated by and assigned against KEGG database[90] using BLASTp (bitscore ≥50, $e \le 10^{-5}$) to verify their accuracy. Functional genes (up to 5 for each viral AMG) from MAGs and eggNOG v5.0.0 database[88] were used for BLASTp-based comparison with these C-fixation AMGs (bitscore ≥ 50, $e \le 10^{-3}$)[85]. For each group of viral AMGs, alignment was performed with MAFFT[93] and columns containing over 95% gaps were eliminated through filtering with TrimAL[94]. Phylogenetic trees were then constructed in Fasttree[95] with default parameters, followed by submitted to iTOL (version 3) for graphical representation and formatting adjustments. In addition, HGT among phylum-level MAGs was investigated using MetaCHIP v1.10.10[96] with default parameters, employing BLASTn best-match and phylogenetic strategies. Structural homology of AMGs were searched using the Phyre2[97] and Sigma-70 transcriptional promoter was identified by SAPPHIRE[98] ($p < 0.05$, https://sapphire.biw.kuleuven.be/index.php).

## Functionally characterization of viral C-related AMGs

The genes encoding for AMGs (i.e., *rbcL1*, *ppdK1*, and *TKT1*) involved in C fixation were synthetized with the pET28a plasmid serving as the cloning vector (Tsingke Biotechnology, Guangzhou, China). Following PCR amplification, *ppdK1* and *TKT1* were inserted into the pCold TF vector (Takara Bio, Shanghai, China). The recombinant plasmids were transfected into *E. coli* BL21 (DE3) cells. Selection of transformants was performed on LB medium containing ampicillin (or kanamycin), with confirmation by PCR and DNA sequencing. *E. coli* BL21 (DE3) cells carrying AMGs expression vector were cultured in LB broth containing 50 µg/mL ampicillin (or kanamycin) at 37 °C. Up the $OD_{600}$ reaching 0.6, 0.4 mM isopropyl β-D-1-thiogalactopyranoside (IPTG) was added to induce AMGs expression and incubated at 15 °C (or 37 °C) for 16 h. After centrifugation, the collected cells were re-suspended with a binding buffer (500 mM NaCl, 10% glycerin, and 20 mM Tris-HCl (pH 8.0)), and treated with on-ice ultrasound. For the *rbcL1* gene with *rbcS*, the bacterial cells were directly lysed by sonication, and the supernatant was collected by centrifugation for subsequent analysis. TF-tagged AMGs (*ppdK1* and *TKT1*) were purified using BeyoGold™ His-tag Purification Resin (Beyotime, Shanghai, China) following the manufacturer's protocol. The protein purification was visually confirmed by SDS-PAGE and Coomassie Brilliant Blue R-250, and the purified protein concentration was measured using the Nanodrop 2000 spectrophotometer (Thermo Scientific, Waltham, USA). The activities of AMGs were analyzed using a double antibody one-step sandwich enzyme-linked immunosorbent assay (ELISA) kit (Shanghai Huding Biotechnology Co., Ltd, Shanghai, China).

## Phage-mediated gene transfer experiment

The recombinant plasmids with C-fixation AMGs (i.e., *rbcL1*, *ppdK1*, and *TKT1*) were transfected into *E. coli* TG1 cells. The helper phage VCSM13 and the host strain *E. coli* TG1 were co-cultured using the double-layer agar method[99]. Plaques were picked and inoculated into 2 × YT broth containing appropriate antibiotics (ampicillin or kanamycin) to amplify the phage, followed by extraction of the phage suspension. The obtained phage suspension was then mixed with the host strain *E. coli* XL1-Blue at a ratio simulating the actual abundance of phages ($\sim 10^9$ pfu/mL) and bacteria ($\sim 10^8$ cfu/mL) in natural environments. The culture was serially diluted, spread onto selective antibiotic plates, and colonies were picked for PCR verification to exclude contamination from the host strain *E. coli* TG1. Primers were specifically designed for *E. coli* TG1, *E. coli* XL1-Blue, plasmid pET30a and plasmid pCold on NCBI website, respectively (Supplementary Data 15). To ensure sufficient substrate concentration for *rbcL1* and *rbcS* expression, ribulose 1,5-bisphosphate (RuBP) was added to *E. coli* XL1-Blue culture ($OD_{600} = 0.6$) to a final concentration of 10 mM. The mixture was incubated on ice for 5–10 min to allow substrate equilibration, followed by electroporation using a 0.1-cm cuvette at 1.8–2.5 kV (adjusted

for strain sensitivity). Immediately after pulsing, cells were recovered in LB medium at 37 °C for 1 h to restore membrane integrity and facilitate intracellular uptake of RuBP. The expression was resuscitated and induced with 0.5 mM IPTG inducer added. Samples were taken at time intervals to detect cell concentrations and $CO_2$ contents.

## Viral inoculation experiment

The soil collected for virus extraction and microcosm incubation was from the abandoned land of Dabaoshan mining areas. Soil virus suspension was extracted from 1 kg of soil and mixed with 1 L of potassium citrate buffer (10.0 g/L $C_6H_5K_3O_7$, 1.92 g/L $Na_2HPO_4 \cdot 12H_2O$, and 0.24 g/L $KH_2PO_4$, pH = 7) at 4 °C for 2 h under 250 rpm and centrifuged at 4 °C for 20 min (5000 × g). The supernatant obtained was filtered using a 2 μm cellulose acetate filter and subsequently incubated with the addition of mitomycin-C (at a final concentration of 1 μg/mL) for 24 h under 200 rpm in the dark under room temperature conditions. The suspensions were taken and passed through 0.22 μm filtration membrane under vacuum at 4 °C to remove bacterial residues. The virus suspension <0.22 μm was then concentrated to 10 mL (100-fold concentration) using tangential flow filtration with a 100 kDa pore size cutoff. Half of the phage concentrate was inactivated after high temperature and autoclave sterilization, and the other half was temporarily stored at 4 °C. In microcosm incubation, each system comprised 20 g of original fresh soil, wherein additional 0.5 g C/kg by $^{13}C$-labeled $NaHCO_3$ was added and 200 μL of active or inactive virus concentrate was inoculated, respectively. The contents of SOC were measured with the same method as described above. The contents of SIC, $^{13}C$-SOC and $CO_2$ were determined using elemental analyzer (Vario EL cube, Germany), isotope ratio mass spectrometers (DELTA V Advantage, USA) and gas chromatograph (GC-2014, China), respectively. Primers were specifically designed for three representative AMGs (*rbcL1*, *ppdK1* and *TKT1*) on NCBI website, respectively (Supplementary Data 15), and their expressions were detected by quantitative real-time PCR (qRT-PCR).

## Transcriptomic analysis of bacterial functional genes

RNA was extracted from 2 g soil samples in microcosm incubation using RNeasy PowerSoil Total RNA Kit on days 0, 3, 7, 14, and 21, and NanoDrop 2000 (Thermo Fisher Scientific, USA) was used to determine purity and concentration. rRNA transcripts were removed using ALFA-SEQ rRNA depletion kit. After the library was qualified, Illumina HiSeq platform was used for paired-end sequencing. Low quality and short reads were filtered with Trimmomatic[72] (LEADING:3 TRAILING:3 SLIDINGWINDOW:5:20 MINLEN:50). For the removal of rRNA reads, SortMeRNA was used to perform alignment of reads against the SILVA SSU (16S/18S) and SILVA LSU (23S/28S) databases[100]. Clean reads were de novo assembled by IDBA_tran (--pre_correction --mink 20 --maxk 60 --step 10)[101]. Prodigal v2.6.3[83] was used to predict ORFs for assembled scaffolds over 500 bp in length, and the unigenes were clustered using Linclust (Default parameters -e 0.001 --min-seq-id 0.95 -c 0.90)[102]. Clean reads were mapped to unigenes with Bowtie2 v2.33[103] under default settings to quantify gene expression levels. The relative abundance of genes was normalized by FPKM using RSEM[104]. BLASTP v2.2.31 was used to align unigenes with the NCBI-NR database for species homology. DIAMOND was used to compare the unigenes with eggNOG[88], CAZy[105] and KEGG[90] databases for functional annotation.

## Statistical analyses

Statistical analyses and data visualization were performed using packages in the statistical program R v4.2.2. The "diversity" function in vegan v2.6-4 was used to examine the alpha diversity index of microorganisms per sample, and the package's "decostand" function (with the "Hellinger" method) was used to standardize the viral and bacterial communities. For community structure analysis, the Bray-Curtis dissimilarity metric (via vegan v2.6-4) was employed to quantify differences, and Mantel tests (via vegan v2.6-4) were conducted to explore correlations between dissimilarity matrices. ANOSIM analysis was used to determine the significant difference of protease activity among different controls. The 'rcorr' function (with 999 permutations) in Hmisc v5.1-2 was applied to compute Pearson correlations, which served to evaluate interactions between the abundances and richness of environmental factors, prokaryotes and viruses. The two-sided *t*-test was used to evaluate the statistical significance between variations.

## Reporting summary

Further information on research design is available in the Nature Portfolio Reporting Summary linked to this article.

## Data availability

The virus sequences generated from contaminated and non-contaminated soil metagenomes in this study have been deposited in the NCBI BioProject database under the accession number PRJNA1196805 and PRJNA1196813, respectively. The soil metagenomic data used in this study are available in the NCBI BioProject database under the accession number PRJNA1253350, PRJNA1253357, PRJNA1253358, PRJNA1253360, and PRJNA1254387. The metatranscriptomic dataset generated from microcosm experiment have been deposited in the NCBI BioProject database under the accession number PRJNA1203377, PRJNA1203296, PRJNA1203294, PRJNA1203300, and PRJNA1203232. The supplementary and source data generated in this study are provided in the Supplementary Information/Source Data file, which have been deposited in the Figshare database [https://doi.org/10.6084/m9.figshare.29924471]. Source data are provided with this paper.

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

## Acknowledgements

Y. Chao was supported by the National Natural Science Foundation of China (No. 42077105 and No. U23A20679), the Guangdong Basic and Applied Basic Research Foundation (No. 2022A1515012028 and No. 2025A1515010760) and the Guangdong Special Support Program (No. 2021TQ060211). Y.T. was supported by the Chinese National Key Research and Development Program of China (No. 2023YFC3905800) and the National Natural Science Foundation of China (No. U22A20605). R.Q. was supported by the 111 Project (No. B18060). The authors thank Dr. Yanmei Chen (School of Life Sciences and Human Phenome Institute, Fudan University, Shanghai, China), Dr. Tao Jin (Guangdong Magigene Biotechnology Co. Ltd., China), Dr. Qicheng Xu (Jiangsu Provincial Key Lab for Solid Organic Waste Utilization, Nanjing Agricultural University, Nanjing, China) and Dr. Yingjie Cao (School of Environmental Science and Engineering, Sun Yat-sen University, Guangzhou, China) for helpful discussions.

## Author contributions

J.L., Y. Chao and R.Q. designed the research. J.L., X.Z. and Z.C. conducted the experiments and collected the data. J.L., L.T., X.Z., B.H. and H.H. analyzed the metagenomic data. J.L., Y. Cao, H.F., S.Z., H.Q. and Yulu Y. conducted the molecular biology experiments. J.L., L.Z. and K.Y. conducted the phage-mediated gene transfer experiments. J.L., C.H., M.L. and Z.F. conducted the viral community inoculation experiments. J.L. and Y.Chao wrote the initial draft of the manuscript while Z.R., K.D., Ying Y., K.Y., W.L., Y.F., S.W., Y.T., N.L., T.L. and Z.X. provided substantial feedback.

## Competing interests

The authors declare no competing interests.
