## [Transparent Peer Review file · Nature Communications]

DNA viral community enhances microbial carbon fixation capacity via auxiliary metabolic genes in contaminated soils

Corresponding Author: Professor Yuanqing Chao

Version 0:

Reviewer comments:

Reviewer #1

(Remarks to the Author)

The authors of this study explored the impact of viruses on carbon fixation in contaminated soils from mining activities using multiomics and large-scale spatial datasets together with experimental results. The results from this study support the viral contributions for carbon fixation and the roles of virus in carbon cycling. It's the first study to explore the positive impact of viruses on carbon fixation in contaminated soils using multiomics and validated experiments.

The methodology in this study is sound but there're a few points that require further explanation;

1). Line 419, DNA extraction and metagenomic sequencing.

The authors extracted DNA from soil samples to analysis viral and bacterial sequences. What about RNA virus ? Since this study is based on virome. What about RNA virus ? Are they included in this study ?

2). Line 811, Fig. 3 Function and environmental distribution of carbon fixation AMGs

The Fig 3c indicates horizontal gene transfer and use grey lines to visualise the event in the figure.

How did the authors define or identify the predicted horizontal gene transfer events (grey lines)? This should be written further in method section.

3). In this study, the authors collected soils samples from abandoned mines, tailing areas, downstream contaminated farmlands and non-contaminated sites, making at total of 409 samples. How many of these samples that the authors extracted DNA and sequenced ?

The author mentioned that only the virus sequences generated from contaminated and non-contaminated soil metagenomes in this study have been deposited in the NCBI BioProject database under the accession number PRJNA1196805 and PRJNA1196813, respectively.

The soil metagenomic samples used in this study, they should be all submitted to ENA for reproducibility and their ENA numbers should be provided in supplementary table 1.

Reviewer #2

(Remarks to the Author)

The manuscript "Virome enhances carbon fixation capacity of microbial community via auxiliary metabolic genes in contaminated soils" by Lu and co-workers deals with the impact of viruses carrying carbon fixation genes in contaminated soils across the Chinese republic. The authors recovered 31,210 viral OTUs, identified 23 AMGs described to be involved in all carbon fixation pathways described so far, and described their effect on carbon fixation potential in soil via a) biochemical

characterization of selected AMG-encoded enzymes and b) mesocosms inoculated with viruses and measurements of gene abundances. I particularly enjoyed the combination of bioinformatics, biochemistry and enrichment cultures. While I do think that the story has merit, the story suffers from some substantial shortcomings that need to be addressed in my honest opinion. These major issues mostly relate to the fact that a) the discovery of AMGs related to CO₂ fixation is not novel and b) that the authors provide no solid evidence that viruses and their AMGs really modulate CO₂ fixation in soil (they provide rather hints than evidence).

Major issues:

1. There are seven reported carbon fixation pathways in the literature, three of which do not have enzymes that are exclusive to respective pathway. For instance, the reverse oxidative TCA has all enzymes that are used for a regular TCA – so how can the authors bioinformatically conclude that enzymes involved in carbon fixation for that particular pathway are encoded in the 23 AMGs? Please also note that ATP-citrate lyase of the reverse TCA (not roTCA!) also occurs in Eukaryotes which are non-carbon fixing (apart from anaerobic reactions). Hence, I do not agree with the statement by the authors that they identified AMGs involved in all known carbon fixation pathways.
2. The number of AMGs / viruses potentially involved in carbon fixation is relatively low compared to the overall number of vOTUs (31,210 vs 23, i.e. 0.07%). I think the authors must flesh out the importance of these AMGs/ vOUTs in the respective ecosystems by also looking at dynamics across space (and time) with respect to their relative abundance.
3. The mesocosm studies are great to measure if the viruses potentially have an impact on the carbon turnover in soil, yet the main experiment is missing. The authors should measure carbon dioxide turnover in these enrichments (maybe also via ¹³C carbon dioxide to measure incorporation rate and targets) to really get to the core what the effect of these AMGs really is. It will be tough to relate the turnover rates to the viral AMGs (rather than other secondary/indirect effects), yet I think this is an important step forward and would mitigate issue 2 that I mentioned. Otherwise the advance that this study offers is quite limited and all the results are only hints rather than evidence that viruses and their AMGs modulate carbon fixation rates.

Minor comments:

Abstract:

- Line 32: something that remains unclear is a weak motivation for investigation.
- Line 35: “types” of genes is unclear, please rephrase.
- Line 39: please be more precise and state that these were enrichment studied as you simply say “in the contaminated soils”. If these were not enrichment cultures please rephrase so that it is no longer ambiguous.
- Line 42: “supports” consider rephrasing

Introduction:

- Please spell out all abbreviations when used for the first time, such as TKT in line 78.
- Line 94: “the microcosm” -> “microcosm experiments”
- Line 95: “necessary” -> “necessity”
- Line 96: “the ecosystems” drop the “the”

Results:

- Line 132-136: The fact that the niches of viruses are much narrower than those of the corresponding hosts has been known for a long time. Please consider citing according literature here (or in the discussion).
- Line 170: “In contrast,” -> “By contrast,” (as compared to “In contrast to XY,”))
- Line 183: “did not correlate exactly” – this is ambiguous, rephrase!
- Line 184: I cannot follow the logic here, i.e. how a missing correlation can suggest HGT.
- Line 187: “between MAGs at the phylum level” this is as clear as mud. Do you mean that there is HGT of MAGs within one phylum or across phyla?
- Line 189: “exactly occurred” – I cannot comprehend what the authors are trying to tell here.
- Line 190: “C-fixation AMGs” consider rephrasing
- Line 191: virus-encoded AMGs – AMGs are by definition always encoded by viral genomes. I think this new terminology is confusing.
- Line 195: does that mean that the authors identified only 23 viral genomes carrying genes related to pathways for carbon fixation?
- Line 222: higher than what?

Discussion:

The discussion is generally way too long. Repetitions of results should be avoided.

- Line 272: Rephrase “in the entire ecosystem” as the reader does not know which ecosystem you are talking about
- Line 278: Replace “habitats” with “ecosystems”.

Version 1:

Reviewer comments:

Reviewer #1

(Remarks to the Author)

The authors have responded and revised the manuscript according to my comments and suggestions. I have no further comment for them.

Reviewer #2

(Remarks to the Author)

Re-reviewing the manuscript by Lu and co-workers, I need to say that it has substantially improved compared to the previous version that I had read. Regarding my queries, most of them were answered satisfyingly. I particularly like that the authors redid their mesocosm experiment with ¹³C labels, which substantially improved the study. Yet, my first major issue was not properly answered and highlights the general issue of the manuscript, i.e. that the interpretation of the results is geared towards supporting the main claim (viruses bolster C-fixation in soil via AMGs) and alternative explanations are not considered.

My major criticism:

"1. There are seven reported carbon fixation pathways in the literature, three of which do not have enzymes that are exclusive to respective pathway. For instance, the reverse oxidative TCA has all enzymes that are used for a regular TCA – so how can the authors bioinformatically conclude that enzymes involved in carbon fixation for that particular pathway are encoded in the 23 AMGs? Please also note that ATP-citrate lyase of the reverse TCA (not roTCA!) also occurs in Eukaryotes which are non-carbon fixing (apart from anaplerotic reactions). Hence, I do not agree with the statement by the authors that they identified AMGs involved in all known carbon fixation pathways."

Authors' answer:

"R: Thanks for the comments. We totally agree that several enzymes involved in roTCA are shared with the regular TCA cycle, which introduces the possibility that some AMGs (i.e. ACO and frdA) might be associated with TCA processes rather than carbon fixation. However, besides ACO and frdA, our study also identified two key genes — korB (encoding the β -subunit of 2-oxoglutarate: ferredoxin oxidoreductase) and ppdK (encoding pyruvate phosphate dikinase) — that are directly involved in the roTCA cycle for carbon fixation. The presence of these roTCA-specific AMGs provides strong evidences supporting the involvement of viruses in promoting carbon fixation through the roTCA pathway.

We note that viral contributions to host regular TCA activity and overall microbial metabolic capabilities remain underexplored. Inspired by your comments, we propose that future studies should investigate the potential roles of viruses not only in carbon fixation but also in broader microbial metabolic processes, including the TCA cycle."

Here are the underlying issues:

- a) The roTCA does not only share several (as the authors state) but all enzymes with the TCA cycle making it impossible to detect via genome/gene analyses (see doi: 10.1038/s41586-021-03456-9).
- b) KOR does not only operate in rTCA/roTCA but also Glu biosynthesis in many anaerobes. In some prokaryotes, this enzyme exhibits strong promiscuity.
- c) FRD can also occur in fumarat respiration.
- d) PpdK is also not specific for autotrophy.

I could continue like this and make statements that the detected AMGs could also occur in many other reactions that have been described in the literature, yet the authors allow only one possible explanation, i.e. involvement in autotrophy. While their ¹³C experiments do support at their claims at least in part, I think the authors should include a section in the manuscript where they describe/discuss other explanations for the identified enzymes and their involvement in other metabolic pathways. It is very likely that not all of these enzymes contribute to carbon fixation and that at least one is bolstering another metabolic pathway.

Version 2:

Reviewer comments:

Reviewer #2

(Remarks to the Author)

The authors have answered all my queries.

I have no more reservations and recommend acceptance of the manuscript. I congratulate the authors to this impressive piece of work.

Responses to Comments from Reviewers

Reviewer #1 (Remarks to the Author):

The authors of this study explored the impact of viruses on carbon fixation in contaminated soils from mining activities using multiomics and large-scale spatial datasets together with experimental results. The results from this study support the viral contributions for carbon fixation and the roles of virus in carbon cycling. It's the first study to explore the positive impact of viruses on carbon fixation in contaminated soils using multiomics and validated experiments.

R: Thank you for the positive comments and encouragement. We further improved the manuscript according to your suggestions.

The methodology in this study is sound but there're a few points that require further explanation;

1). Line 419, DNA extraction and metagenomic sequencing.

The authors extracted DNA from soil samples to analysis viral and bacterial sequences. What about RNA virus? Since this study is based on virome. What about RNA virus? Are they included in this study?

R: Thanks for the comments. We appreciate your valuable feedback regarding the inclusion of RNA viruses in our study. In the present study, RNA viruses were not included in the present study as our DNA extraction and metagenomic sequencing protocols were specifically designed to target DNA-based viruses and bacterial sequences. However, as pointed out, RNA viruses are an important component of the virome, and their inclusion would be crucial for a comprehensive understanding. Given the difficulty of RNA extraction for large scale soil samples and the current limitations in the annotation and comprehensiveness of RNA virus databases, the present study only focused on the DNA viruses and their potential contributions on soil carbon fixation. Future studies would benefit from integrating RNA virus analysis. To clarify this and avoid the potential misunderstanding, we revised the title of manuscript as “DNA virome enhances carbon fixation capacity of microbial community via auxiliary metabolic genes in contaminated soils”. Thanks again for your valuable suggestion.

2). Line 811, Fig. 3 Function and environmental distribution of carbon fixation AMGs
The Fig 3c indicates horizontal gene transfer and use grey lines to visualize the event in the figure.

How did the authors define or identify the predicted horizontal gene transfer events (grey lines)? This should be written further in method section.

R: Thanks for the comments. In our study, horizontal gene transfer events were predicted based on sequence similarity and phylogenetic analysis. Specifically, we used MetaCHIP v1.10.10 to identify gene sequences that showed evidence of acquisition from different taxa, indicating a potential HGT event. The grey lines in Fig. 3c were used to visualize these predicted HGT events among MAGs at the phylum level, based on sequence alignment and phylogenetic analysis.

We revised the methods section of the manuscript to include a more detailed description of how these HGT events were predicted. **Please refer to Line 497-499:** *“In addition, HGT among MAGs at the phylum level was explored by MetaCHIP v1.10.10⁹¹ with default parameters using best-match (BLASTn) and phylogenetic approaches.”*

3). In this study, the authors collected soils samples from abandoned mines, tailing areas, downstream contaminated farmlands and non-contaminated sites, making a total of 409 samples. How many of these samples that the authors extracted DNA and sequenced?

R: Thanks for the comments. In this study, DNA extraction and subsequent sequencing were performed on all 409 soil samples collected. **We revised the manuscript to include this information for greater clarity (Line 436-438) as follows.**

“Finally, purified DNA of 409 soil samples were sequenced on the Illumina NovaSeq platform (350 bp insert size, paired-end 150 bp) by Novogene (Beijing, China).”

The author mentioned that only the virus sequences generated from contaminated and non-contaminated soil metagenomes in this study have been deposited in the NCBI BioProject database under the accession number PRJNA1196805 and PRJNA1196813, respectively.

The soil metagenomic samples used in this study, they should be all submitted to ENA for reproducibility and their ENA numbers should be provided in supplementary table 1.

R: Thanks for the comments. The soil metagenomic samples used in this study have been deposited in the NCBI BioProject database under the accession number

PRJNA1253350, PRJNA1253357, PRJNA1253358, PRJNA1253360 and PRJNA1254387. **And the accession numbers were also provided in the revised manuscript (Line 920-923) as follows.**

“The soil metagenomic data used in this study has been deposited in the NCBI BioProject database under the accession number PRJNA1253350, PRJNA1253357, PRJNA1253358, PRJNA1253360 and PRJNA1254387.”

Reviewer #2 (Remarks to the Author):

The manuscript “Virome enhances carbon fixation capacity of microbial community via auxiliary metabolic genes in contaminated soils” by Lu and co-workers deals with the impact of viruses carrying carbon fixation genes in contaminated soils across the Chinese republic. The authors recovered 31,210 viral OTUs, identified 23 AMGs described to be involved in all carbon fixation pathways described so far, and described their effect on carbon fixation potential in soil via a) biochemical characterization of selected AMG-encoded enzymes and b) mesocosms inoculated with viruses and measurements of gene abundances. I particularly enjoyed the combination of bioinformatics, biochemistry and enrichment cultures.

While I do think that the story has merit, the story suffers from some substantial shortcomings that need to be addressed in my honest opinion. These major issues mostly relate to the fact that a) the discovery of AMG related to CO₂ fixation is not novel and b) that the authors provide no solid evidence that viruses and their AMGs really modulate CO₂ fixation in soil (they provide rather hints than evidence).

R: Thanks for the comments. We greatly appreciate your positive feedback on our study. We also thank you for your constructive critique regarding to the novelty and evidence for the role of viruses and AMGs in carbon fixation in soil.

Although several carbon fixation genes have been previously reported in the literature, our study contributed with new insights. Firstly, we expanded the diversity of AMGs related to carbon fixation, 8 categories among which (i.e., *korB*, *frdA*, *ACO*, *GAPDH*, *glpX*, *ACSS*, *fold*, and *MCEE*) were not previously reported in viral AMGs. Secondly, we provided experimental evidence of viral potential role in enhancing microbial carbon fixation through enrichment cultures. These findings collectively highlight the critical roles of viruses in adaptively modulating carbon cycle in response to environmental changes and potentially maintaining carbon reservoirs in stressed ecosystems, particularly in contaminated soils.

Regarding the direct evidence for the modulation of carbon fixation by viruses and their AMGs, we supplemented two investigations, including ¹³C labeled microcosm incubation with quantitative real-time PCR of AMGs and the phage-mediated gene transduction experiment. These experiments confirmed that the C-fixation AMGs could be transduced to bacterial hosts via phage infection, further enhanced the expression of

C-fixation genes, and finally promoted the transformation of inorganic carbon to SOC. We believe these results could consolidate our previous statement of viruses and their AMGs really modulate soil carbon fixation.

For your reference, the methods, results and discussion of ^{13}C labeled microcosm incubation experiment were added on Line 557-566, Line 229-245, Line 348-352 and Fig 5.

On Line 557-566 in the Methods section, we write:

*“In microcosm incubation, each system comprised 20 g of original fresh soil, wherein additional 0.5 g C/kg by ^{13}C -labeled NaHCO_3 was added and 200 μL of active or inactive virus concentrate was inoculated, respectively. The contents of SOC were measured with the same method as described above. The contents of SIC, ^{13}C -SOC and CO_2 were measured using elemental analyzer (Vario EL cube, Germany), isotope ratio mass spectrometers (DELTA V Advantage, USA) and gas chromatograph (GC-2014, China), respectively. Primers were specifically designed for three representative AMGs (*rbcL1*, *ppdK1* and *TKT1*), respectively (Supplementary data 15), and their expressions were detected by quantitative real-time PCR (qRT-PCR).”*

On Line 229-245 in the Results section, we write:

*“For further verification, a viral community inoculation experiment with ^{13}C labeled inorganic C source was conducted to explore the effects of viruses on contaminated soil C fixation. After active virus inoculation, soil inorganic C (SIC) demonstrated a consistent decline (Fig. 5a). Compared to the inactive inoculation, the treatment inoculated with active viruses exhibited a more substantial reduction of SIC (~10%, $p < 0.01$, Fig. 5a). Meanwhile, SOC significantly increased by ~11% ($p < 0.05$), representing a 5.8-fold higher increase than the inactive virus treatment (~1.9%) (Fig. 5b). Notably, active virus inoculation exhibited greater ^{13}C -SOC by approximately ~10% ($p < 0.01$) compared to inactive virus treatment (Fig. 5d), suggesting the enhanced accumulation of SOC was from the labeled inorganic C source. These results provided direct evidences that viruses could significantly promote C fixation. Additionally, it was observed that the expression of *rbcL1*, *ppdK1* and *TKT1* significantly increased after active virus inoculation (Fig. 5e, 5f and 5g). The abundance of C-fixation genes in the active virus treatment was also significantly higher (~73%, $p < 0.05$) than that in the*

inactivated virus treatment (Fig. 5h). Combining the results of soil C turnover and genes expression, the role of viral AMGs in strengthening C fixation was highlighted.”

On Line 348-352 in the Discussion section, we write:

“In the current study, the increase of ^{13}C -SOC provided direct evidence for virus-mediated C fixation (Fig. 5d). In contrast to “Viral Shunt”⁵⁶, the nature of the phenotypic modifications by viruses, such as increased autotrophs abundance (Fig. 6c) and up-regulated microbial C-fixation genes (Fig. 6a), may further directly affect C storage (Fig. 5b)⁵⁷.”

On Fig 5, we write:

Fig. 5 Temporal dynamics of soil carbon turnover, AMGs, and microbial functional genes after active and inactive virus inoculation. Changes of SOC (a), SIC (b), CO₂ (c) and ^{13}C -SOC (d) after viral inoculation. Variation of AMGs, including *rbcL1* (e), *ppdK1* (f) and *TKT1* (g), and microbial C-fixation genes (h) after viral inoculation. Microbial C-fixation genes in each metabolic pathway were first averaged to derive the within-pathway mean (e.g., average FPKM for Calvin cycle genes), followed by a secondary averaging of pathway-specific means to mitigate gene count bias.”

And the methods and results of phage-mediated gene transduction were added on Line 525-544, Line 216-224, Fig. 4e and Supplementary Fig. 10.

On Line 525-544 in the Methods section, we write:

“Phage-mediated gene transfer experiment

The recombinant plasmids with C-fixation AMGs (i.e., *rbcL1*, *ppdK1*, and *TKT1*) were

*transfected into E. coli TG1 cells. The helper phage VCSM13 and the host strain E. coli TG1 were co-cultured using the double-layer agar method⁹⁴. Plaques were picked and inoculated into 2×YT broth containing appropriate antibiotics (ampicillin or kanamycin) to amplify the phage, followed by extraction of the phage suspension. The obtained phage suspension was then mixed with the host strain E. coli XL1-Blue at a ratio simulating the actual abundance of phages (~ 10⁹ pfu/mL) and bacteria (~ 10⁸ cfu/mL) in natural environments. The culture was serially diluted, spread onto selective antibiotic plates, and colonies were picked for PCR verification to exclude contamination from the host strain E. coli TG1 (Supplementary data 15). To ensure sufficient substrate concentration for *rbcL1* and *rbcS* expression, ribulose 1,5-bisphosphate (RuBP) was added to E. coli XL1-Blue culture (OD₆₀₀ = 0.6) to a final concentration of 10 mM. The mixture was incubated on ice for 5–10 min to allow substrate equilibration, followed by electroporation using a 0.1-cm cuvette at 1.8–2.5 kV (adjusted for strain sensitivity). Immediately after pulsing, cells were recovered in LB medium at 37°C for 1 h to restore membrane integrity and facilitate intracellular uptake of RuBP. The expression was resuscitated and induced by adding 0.5 mM IPTG inducer. Samples were taken at time intervals to detect cell concentrations and CO₂ contents.”*

On Line 216-224 in the Results section, we write:

*“In addition, using VCSM13 helper phage as a vector, C-fixation AMGs (i.e. *ppdK1*, *TKT1* and *rbcL1*) from E. coli TG1 were successfully transduced into E. coli XL1 Blue. Gene transfer was confirmed by antibiotic screening and PCR amplification (Supplementary Fig. 10). This result highlighted the crucial role of viruses as genetic vector in promoting horizontal transfer of C-fixation genes. Given the active co-expression of *rbcL1* and *rbcS*, E. coli XL1 Blue emitted less CO₂ (Fig. 4e), indicating that the heterotrophic E. coli XL1 Blue obtained C-fixation ability by transferring relevant AMGs.”*

On Fig 4e, we write:

“ Fig. 4e Growth curves of bacteria and the temporal variations of the CO₂. The CO₂ concentration is represented by the difference generated from XL1-Blue with pET30a-rcbL-rcbS and XL1-Blue with pET30a.”

On Supplementary Fig. 10, we write:

“ Supplementary Fig. 10 Results and verification of phage transduction experiments. a

Laques formed by phage infection of host cells. b Bacterial colonies obtained by antibiotic screening after successful phage transduction. c PCR amplification used to verify the exclusion of host contamination.”

We sincerely appreciate the reviewer again for these insightful comments, which would be instrumental in enhancing the quality of the manuscript.

Major issues:

1. There are seven reported carbon fixation pathways in the literature, three of which do not have enzymes that are exclusive to respective pathway. For instance, the reverse oxidative TCA has all enzymes that are used for a regular TCA – so how can the authors bioinformatically conclude that enzymes involved in carbon fixation for that particular pathway are encoded in the 23 AMGs? Please also note that ATP-citrate lyase of the reverse TCA (not roTCA!) also occurs in Eukaryotes which are non-carbon fixing (apart from anaplerotic reactions). Hence, I do not agree with the statement by the authors that they identified AMGs involved in all known carbon fixation pathways.

R: Thanks for the comments. We totally agree that several enzymes involved in roTCA are shared with the regular TCA cycle, which introduces the possibility that some AMGs (i.e. *ACO* and *frdA*) might be associated with TCA processes rather than carbon fixation. However, besides *ACO* and *frdA*, our study also identified two key genes — *korB* (encoding the β -subunit of 2-oxoglutarate: ferredoxin oxidoreductase) and *ppdK* (encoding pyruvate phosphate dikinase) — that are directly involved in the roTCA cycle for carbon fixation. The presence of these roTCA-specific AMGs provides strong evidences supporting the involvement of viruses in promoting carbon fixation through the roTCA pathway.

We note that viral contributions to host regular TCA activity and overall microbial metabolic capabilities remain underexplored. Inspired by your comments, we propose that future studies should investigate the potential roles of viruses not only in carbon fixation but also in broader microbial metabolic processes, including the TCA cycle.

2. The number of AMGs / viruses potentially involved in carbon fixation is relatively low compared to the overall number of vOTUs (31,210 vs 23, i.e. 0.07%). I think the authors must flesh out the importance of these AMGs/vOTUs in the respective

ecosystems by also looking at dynamics across space (and time) with respect to their relative abundance.

R: Thanks for the comments. We would like to point out that, according to the KEGG database, genes involved in carbon fixation (112 independent KOs) account for only approximately 0.4% of all 27,135 independent KOs. This highlights that carbon fixation pathways constitute a very small fraction of overall metabolic potential in microbial communities. Consequently, the inherently low abundance of carbon fixation-related AMGs among viral genomes is expected and could partially reflect the natural distribution of functional genes.

Furthermore, we compared our findings with several recent studies investigating viral AMGs across different biogeochemical processes (Kieft et al. 2021; Liang et al. 2024; Graham et al. 2024). Consistently, viral AMGs related to sulfur (0.031%), phosphorus (0.295%) and carbon cycling (0.181%) occurred at relatively low proportions. Importantly, rare functional genes or organisms can have keystone roles in ecosystem processes. Despite the relatively low proportion of carbon fixation AMGs detected in our study, their ecological contributions on microbial carbon fixation and organic carbon accumulation could not be diminished. This could be further confirmed by the ^{13}C labeled microcosm incubation (Fig. 5), in which the expression of C-fixation AMGs in the active virus treatment significantly increased, and the transformation of inorganic carbon to SOC was effectively promoted. **These results were added on Line 229-245 as follows.** Thanks again for these inspiring comments.

“For further verification, a viral community inoculation experiment with ^{13}C labeled inorganic C source was conducted to explore the effects of viruses on contaminated soil C fixation. After active virus inoculation, soil inorganic C (SIC) demonstrated a consistent decline (Fig. 5a). Compared to the inactive inoculation, the treatment inoculated with active viruses exhibited a more substantial reduction of SIC (~10%, $p < 0.01$, Fig. 5a). Meanwhile, SOC significantly increased by ~11% ($p < 0.05$), representing a 5.8-fold higher increase than the inactive virus treatment (~1.9%) (Fig. 5b). Notably, active virus inoculation exhibited greater ^{13}C -SOC by approximately ~10% ($p < 0.01$) compared to inactive virus treatment (Fig. 5d), suggesting the enhanced accumulation of SOC was from the labeled inorganic C source. These results provided direct evidences that viruses could significantly promote C fixation. Additionally, it was

observed that the expression of *rbcL1*, *ppdK1* and *TKT1* significantly increased after active virus inoculation (Fig. 5e, 5f and 5g). The abundance of C-fixation genes in the active virus treatment was also significantly higher (~73%, $p < 0.05$) than that in the inactivated virus treatment (Fig. 5h). Combining the results of soil C turnover and genes expression, the role of viral AMGs in strengthening C fixation was highlighted.”

References:

Kieft, K., Zhou, Z., Anderson, R.E. et al. Ecology of inorganic sulfur auxiliary metabolism in widespread bacteriophages. *Nature Communications* 12, 3503 (2021).

Liang, J.L., Feng, S.W., Lu, J.L. et al. Hidden diversity and potential ecological function of phosphorus acquisition genes in widespread terrestrial bacteriophages. *Nature Communications* 15, 2827 (2024).

Graham, E.B., Camargo, A.P., Wu, R. et al. A global atlas of soil viruses reveals unexplored biodiversity and potential biogeochemical impacts. *Nature Microbiology* 9, 1873–1883 (2024).

3. The mesocosm studies are great to measure if the viruses potentially have an impact on the carbon turnover in soil, yet the main experiment is missing. The authors should measure carbon dioxide turnover in these enrichments (maybe also via ^{13}C carbon dioxide to measure incorporation rate and targets) to really get to the core what the effect of these AMGs really is. It will be tough to relate the turnover rates to the viral AMGs (rather than other secondary/indirect effects), yet I think this is an important step forward and would mitigate issue 2 that I mentioned. Otherwise the advance that this study offers is quite limited and all the results are only hints rather than evidence that viruses and their AMGs modulate carbon fixation rates.

R: Thanks for the comments. As suggested, the microcosm incubation experiment using ^{13}C labeled inorganic carbon was supplemented. It was found viral-infected microbial communities exhibited 7.2–11.9% greater ^{13}C fixation, and viral AMGs exhibited higher gene expression (i.e. *rbcL1* by 140%, *ppdK1* by 462% and *TKT1* by 239%), providing direct evidence that viral activity enhances carbon fixation efficiency via AMGs. **This information was added in the revised manuscript (Line 229-245 and Fig. 5d-g) as follows.**

“For further verification, a viral community inoculation experiment with ^{13}C labeled

inorganic C source was conducted to explore the effects of viruses on contaminated soil C fixation. After active virus inoculation, soil inorganic C (SIC) demonstrated a consistent decline (Fig. 5a). Compared to the inactive inoculation, the treatment inoculated with active viruses exhibited a more substantial reduction of SIC (~10%, $p < 0.01$, Fig. 5a). Meanwhile, SOC significantly increased by ~11% ($p < 0.05$), representing a 5.8-fold higher increase than the inactive virus treatment (~1.9%) (Fig. 5b). Notably, active virus inoculation exhibited greater ^{13}C -SOC by approximately ~10% ($p < 0.01$) compared to inactive virus treatment (Fig. 5d), suggesting the enhanced accumulation of SOC was from the labeled inorganic C source. These results provided direct evidences that viruses could significantly promote C fixation. Additionally, it was observed that the expression of *rbcL1*, *ppdK1* and *TKT1* significantly increased after active virus inoculation (Fig. 5e, 5f and 5g). The abundance of C-fixation genes in the active virus treatment was also significantly higher (~73%, $p < 0.05$) than that in the inactivated virus treatment (Fig. 5h). Combining the results of soil C turnover and genes expression, the role of viral AMGs in strengthening C fixation was highlighted.”

Fig. 5 Temporal dynamics of soil carbon turnover, AMGs, and microbial functional genes after active and inactive virus inoculation. Changes of SOC (a), SIC (b), CO_2 (c) and ^{13}C -SOC (d) after viral inoculation. Variation of AMGs, including *rbcL1* (e), *ppdK1* (f) and *TKT1* (g), and microbial C-fixation genes (h) after viral inoculation. Microbial C-fixation genes in each metabolic pathway were first averaged to derive the within-pathway mean (e.g., average FPKM for Calvin cycle genes), followed by a secondary averaging of pathway-specific means to mitigate gene count bias.”

In addition, using VCXM13 helper phage as a vector (Supplementary Fig. 10), C-

fixation AMGs (i.e. *rbcL1*, *ppdK1* and *TKT1*) from *E. coli* TG1 were successfully transduced into *E. coli* XL1 Blue and effectively promoted the carbon fixation ability of the host (Fig. 4e). These results further highlighted the potential roles of DNA virome in mediating soil carbon fixation through horizontal transfer of AMGs. **These results were added in the revised manuscript (Line 216-224, Fig 4e and Supplementary Fig. 10) as follows.**

“In addition, using VCSM13 helper phage as a vector, C-fixation AMGs (i.e. *ppdK1*, *TKT1* and *rbcL1*) from *E. coli* TG1 were successfully transduced into *E. coli* XL1 Blue. Gene transfer was confirmed by antibiotic screening and PCR amplification (Supplementary Fig. 10). This result highlighted the crucial role of viruses as genetic vector in promoting horizontal transfer of C-fixation genes. Given the active co-expression of *rbcL1* and *rbcS*, *E. coli* XL1 Blue emitted less CO₂ (Fig. 4e), indicating that the heterotrophic *E. coli* XL1 Blue obtained C-fixation ability by transferring relevant AMGs.”

“ Fig. 4e Growth curves of bacteria and the temporal variations of the CO₂. The CO₂ concentration is represented by the difference generated from XL1-Blue with pET30a-rbcL-rbcS and XL1-Blue with pET30a.”

“*Supplementary Fig. 10 Results and verification of phage transduction experiments. a Laques formed by phage infection of host cells. b Bacterial colonies obtained by antibiotic screening after successful phage transduction. c PCR amplification used to verify the exclusion of host contamination.*”

To sum up, we tried our best to seek the direct evidence that viruses and their AMGs modulate carbon fixation. Hopefully, it would help to reinforce the supports to our discovery. We would like to thank the reviewer again for providing these important and constructive comments.

Minor comments:

Abstract:

- Line 32: something that remains unclear is a weak motivation for investigation.

R: Thanks for the comment. We revised the abstract to better articulate the motivation for the present investigation. **Please refer to Line 31-34 as follows.**

“While a few carbon-related auxiliary metabolic genes (AMGs) in viruses were reported to potentially influence the hosts, the effects of virus-host interactions on the soil carbon fixation, particularly in carbon-deficient contaminated soils, needed further validation.”

- Line 35: “types” of genes is unclear, please rephrase.

R: Thanks for the comment. The phrase “types of genes” has been rephrased to “functional categories of carbon fixation-related auxiliary metabolic genes (AMGs)” for better precision and clarity. **Please refer to Line 36 as follows.**

“Eleven different functional categories of carbon fixation AMGs, covering all the currently known carbon fixation pathways, were identified in the contaminated soils for the first time.”

- Line 39: please be more precise and state that these were enrichment studied as you simply say “in the contaminated soils”. If these were not enrichment cultures please rephrase so that it is no longer ambiguous.

R: Thanks for the comment. As suggested, we clarified that the experimental setup involved enrichment cultures derived from contaminated soils rather than direct measurements in bulk soil. **Please refer to Line 42-43 as follows.**

“Furthermore, active virus inoculation significantly up-regulated the carbon fixation genes (~73%, $p < 0.05$) and significantly increased the accumulation of organic carbon (~11%, $p < 0.05$) and ^{13}C -labeled organic carbon (~10%, $p < 0.01$) in enrichment cultures derived from contaminated soils.”

- Line 42: “supports” consider rephrasing

R: Thanks for the comment. We replaced “supports” by “evidences” as suggested.

Please refer to Line 44 as follows.

“Our results provide theoretical and experimental evidences for incorporating viral contributions into the assessments of carbon fixation, and complete the roles of virus within the whole processes of carbon cycling.”

Introduction:

- Please spell out all abbreviations when used for the first time, such as TKT in line 78.

R: Thanks for the comment. Revised as suggested, **please refer to Line 81, Line 83 and Line 85-86 as follows.**

“For example, in deep sea sediments, viruses encoded AMGs such as transketolase (TKT) and pyruvate orthophosphate dikinase (ppdK) in Calvin Benson [CB] cycle and rTCA cycle, respectively^{25,26}. Viruses in great salt lake contained ribulose-bisphosphate carboxylase large chain (rbcL) involved in CB cycle for C fixation²⁷. In soil ecosystems, virus-mediated potential C-fixation AMGs, including TKT, ribose 5-phosphate isomerase A (RpiA), and ribose-phosphate pyrophosphokinase (PrsA), were also reported²⁸.”

- Line 94: “the microcosm” -> “microcosm experiments”

R: Thanks for the comment. Revised as suggested, **please refer to Line 98.**

“The effects of viral inoculation on the C-fixation function of microbial communities and the changes of SOC were quantified in microcosm experiments.”

- Line 95: “necessary” -> “necessity”

R: Thanks for the comment. Revised as suggested, **please refer to Line 100.**

“Overall, our study provides key insights on the positive effects of viruses on C fixation in the contaminated soils, and reinforces the necessity to incorporate viral contributions into assessments of C fixation across ecosystems.”

- Line 96: “the ecosystems” drop the “the”

R: Thanks for the comment. Revised as suggested, please refer to Line 101.

“Overall, our study provides key insights on the positive effects of viruses on C fixation in the contaminated soils, and reinforces the necessity to incorporate viral contributions into assessments of C fixation across ecosystems.”

Results:

- Line 132-136: The fact that the niches of viruses are much narrower than those of the corresponding hosts has been known for a long time. Please consider citing according literature here (or in the discussion).

R: Thanks for the comment. Revised as suggested. Relevant reference was added in the discussion to explain the narrow and unique ecological niches of viruses. For details, please refer to Line 285-287.

“Few vOTUs and VCs were shared among different ecosystems (Fig. 1e), further revealing a high spatial turnover rate and within viral communities^{28,31} and uniqueness of the viral ecological niche³².”

- Line 170: “In contrast,” -> “By contrast,” (as compared to “In contrast to XY,”)

R: Thanks for the comment. Revised as suggested, please refer to Line 176.

“By contrast, the influence of viruses carrying C-fixation AMGs on SOC was not detected in the non-contaminated soils (Fig. 3b).”

- Line 183: “did not correlate exactly” – this is ambiguous, rephrase!

R: Thanks for the comment. Rephrased as suggested, please refer to Line 189.

“Notably, the phylogeny of these homologs did not correlate well with the taxonomic classification of microbial species (Fig. 3c), suggesting a significant role of horizontal gene transfer (HGT) in shaping their distribution.”

- Line 184: I cannot follow the logic here, i.e. how a missing correlation can suggest HGT.

R: Thanks for the comment. Horizontal gene transfer (HGT) is the most remarkable feature of bacterial evolution, driving bacterial adaptation to new ecological niches through recombination events (Hall et al. 2020; Irwin et al. 2022). The HGT event disrupts phylogenetic congruence, leading to asymmetry between gene phylogeny and species distribution (Jian et al. 2021; Arnold et al. 2022). Thus, the missing correlation could suggest a significant role of HGT in shaping genes distribution. We clarified this in the revised manuscript. **For details, please refer to Line 188-191.**

“Notably, the phylogeny of these homologs did not correlate well with the taxonomic classification of microbial species (Fig. 3c), suggesting a significant role of horizontal gene transfer (HGT) in shaping their distribution.”

References:

Hall R.J., Whelan F.J., McInerney J.O., Ou Y. & Domingo-Sananes M.R. Horizontal gene transfer as a source of conflict and cooperation in prokaryotes. *Frontiers in Microbiology* 11, 1664-302X (2020).

Jian, H. et al. Diversity and distribution of viruses inhabiting the deepest ocean on Earth. *The ISME Journal* 15, 3094-3110 (2021).

Irwin, N.A.T., Pittis, A.A., Richards, T.A. & Patrick J.K. Systematic evaluation of horizontal gene transfer between eukaryotes and viruses. *Nature Microbiology* 7, 327-336 (2022).

Arnold, B.J., Huang, IT. & Hanage, W.P. Horizontal gene transfer and adaptive evolution in bacteria. *Nature Reviews Microbiology* 20, 206-218 (2022).

- Line 187: “between MAGs at the phylum level” this is as clear as mud. Do you mean that there is HGT of MAGs within one phylum or across phyla?

R: Thanks for the comment. The HGTs between MAGs occurred both within one phylum and across phyla. We revised the statement to avoid potential misunderstanding. **For details, please refer to Line 192-193.**

“Overall, there were numerous HGTs between MAGs both within one phylum and across phyla.”

- Line 189: “exactly occurred” – I cannot comprehend what the authors are trying to

tell here.

R: Thanks for the comment. Revised as suggested, please refer to Line 193-196.

“While, specific HGTs of AMGs exactly occurred between MAGs and viruses, e.g., korB transferred from Elusimicrobiota to Mesyanzhinovviridae and ppdK transferred from Vilmaviridae to Planctomycetota (Fig. 3c).”

- Line 190: “C-fixation AMGs” consider rephrasing.

R: Thanks for the comment. Here, the C-fixation AMGs refer to the auxiliary metabolic genes that are associated with carbon fixation processes. The term was defined in Line 79-80.

“Remarkably, lysogenic viruses may have further effects on the soil C cycle by reprogramming host metabolism by encoding C fixation auxiliary metabolic genes (C-fixation AMGs).”

- Line 191: virus-encoded AMGs – AMGs are by definition always encoded by viral genomes. I think this new terminology is confusing.

R: Thanks for the comment. The term of vAMGs was removed to avoid confusing. Please refer to Line 197.

“Most of the C-fixation AMGs were located between two viral marker genes, including rbcL1, ppdK1, and TKT1 (Fig. 4a).”

- Line 195: does that mean that the authors identified only 23 viral genomes carrying genes related to pathways for carbon fixation?

R: Thanks for the comment. Yes, it does, which was clarified in the manuscript, as shown on Line 201.

“Among the 23 viral genomes carrying the C-fixation AMGs, 6 vOTUs were identified as prophages, while 7 vOTUs exhibited lysogenic potential (Fig. 4b).”

- Line 222: higher than what?

R: Thanks for the comment. In the revised manuscript, we supplemented the ¹³C labeled microcosm incubation with quantitative real-time PCR of AMGs, confirming that C-fixation AMGs could promote the transformation of inorganic carbon to SOC. After a second consideration, the correlation analysis of microbial C-fixation genes and SOC was removed from the revised manuscript.

Discussion:

The discussion is generally way too long. Repetitions of results should be avoided.

R: Thanks for the comment. The recalled results in the discussion were as short as possible when necessary. Repetitions were avoided as suggested, **e.g., Line 281-283, Line 320-323, Line 348-352 and so on.**

“About 84% of viruses in the contaminated soils remain highly unknown (Fig. 1d), which is consistent with previous studies on inorganic²⁹ and organic³⁰ contaminated soils.”

“In the present study, autotrophic bacteria (Xanthobacteraceae, Rhodanobacteraceae and Nitrospiraceae) in the soil ecosystem was found to be positively affected by viruses for the first time (Fig. 6c).”

“In the current study, the increase of ¹³C-SOC provided direct evidence for virus-mediated C fixation (Fig. 5d). In contrast to “Viral Shunt”⁵⁶, the nature of the phenotypic modifications by viruses, such as increased autotrophs abundance (Fig. 6c) and up-regulated microbial C-fixation genes (Fig. 6a), may further directly affect C storage (Fig. 5b)⁵⁷.”

- Line 272: Rephrase “in the entire ecosystem” as the reader does not know which ecosystem you are talking about.

R: Thanks for the comment. “In the entire ecosystem” has been revised to specify the exact ecosystem in discussion. **Please refer to Line 290-291.**

“Viral infection is ubiquitous in various ecosystems (e.g., terrestrial and aquatic ecosystems)^{17,18,35}, wherein viral expression of AMGs diverts host metabolisms and could enhance the adaptability of both hosts and viruses in the soil³⁶.”

- Line 278: Replace “habitats” with “ecosystems”.

R: Thanks for the comment. Revised as suggested, **please refer to Line 297.**

“In terrestrial ecosystems, a global atlas of soil viruses revealed that AMGs (i.e. TKT, RpiA and PrsA) related to C fixation might participate in the CB cycle²⁸, but there is a lack of direct evidence to demonstrate the role of AMGs in C fixation.”

Responses to Comments from Reviewers

Reviewer #1 (Remarks to the Author):

The authors have responded and revised the manuscript according to my comments and suggestions. I have no further comment for them.

R: Thank you for raising valuable points that allowed us to refine the details of the manuscript. We sincerely appreciate your thoughtful feedbacks and are grateful for your recognition of our work.

Reviewer #2 (Remarks to the Author):

Re-reviewing the manuscript by Lu and co-workers, I need to say that it has substantially improved compared to the previous version that I had read. Regarding my queries, most of them were answered satisfyingly. I particularly like that the authors redid their mesocosm experiment with ^{13}C labels, which substantially improved the study. Yet, my first major issue was not properly answered and highlights the general issue of the manuscript, i.e. that the interpretation of the results is geared towards supporting the main claim (viruses bolster C-fixation in soil via AMGs) and alternative explanations are not considered.

R: Thank you sincerely for your re-review and recognizing the substantial improvements in our revised manuscript. We totally agree with your critical point that sufficient consideration of alternative explanations should be supplemented. Therefore, we made thorough revisions in Abstract, Results and Discussion (details shown below). Thank you again for your suggestions to strengthen our work.

My major criticism:

“1. There are seven reported carbon fixation pathways in the literature, three of which do not have enzymes that are exclusive to respective pathway. For instance, the reverse oxidative TCA has all enzymes that are used for a regular TCA – so how can the authors bioinformatically conclude that enzymes involved in carbon fixation for that particular pathway are encoded in the 23 AMGs? Please also note that ATP-citrate lyase of the reverse TCA (not roTCA!) also occurs in Eukaryotes which are non-carbon fixing

(apart from anaplerotic reactions). Hence, I do not agree with the statement by the authors that they identified AMGs involved in all known carbon fixation pathways.”

Authors’ answer:

“R: Thanks for the comments. We totally agree that several enzymes involved in roTCA are shared with the regular TCA cycle, which introduces the possibility that some AMGs (i.e. ACO and frdA) might be associated with TCA processes rather than carbon fixation. However, besides ACO and frdA, our study also identified two key genes — korB (encoding the β -subunit of 2-oxoglutarate: ferredoxin oxidoreductase) and ppdK (encoding pyruvate phosphate dikinase) — that are directly involved in the roTCA cycle for carbon fixation. The presence of these roTCA-specific AMGs provides strong evidences supporting the involvement of viruses in promoting carbon fixation through the roTCA pathway.

We note that viral contributions to host regular TCA activity and overall microbial metabolic capabilities remain underexplored. Inspired by your comments, we propose that future studies should investigate the potential roles of viruses not only in carbon fixation but also in broader microbial metabolic processes, including the TCA cycle.”

Here are the underlying issues:

- a) The roTCA does not only share several (as the authors state) but all enzymes with the TCA cycle making it impossible to detect via genome/gene analyses (see doi: 10.1038/s41586-021-03456-9).
- b) KOR does not only operate in rTCA/roTCA but also Glu biosynthesis in many anaerobes. In some prokaryotes, this enzyme exhibits strong promiscuity.
- c) FRD can also occur in fumarat respiration.
- d) PpdK is also not specific for autotrophy.

I could continue like this and make statements that the detected AMGs could also occur in many other reactions that have been described in the literature, yet the authors allow only one possible explanation, i.e. involvement in autotrophy.

While their ^{13}C experiments do support at their claims at least in part, I think the authors should include a section in the manuscript where they describe/discuss other explanations for the identified enzymes and their involvement in other metabolic

pathways. It is very likely that not all of these enzymes contribute to carbon fixation and that at least one is bolstering another metabolic pathway.

R: Thanks for your insightful suggestions regarding the potential multifunctionality of the detected AMGs in the roTCA cycle and the need to consider alternative metabolic pathways beyond autotrophy. We fully acknowledge that some of these identified AMGs involved in TCA/roTCA, including *korB*, *frdA* and *ppdK*, have well-documented roles in several different metabolic pathways, indicating that the previous statement of those AMGs was inaccurate. To address this, we removed the statement that the identified AMGs covered all currently known microbial C-fixation pathways, please refer to Line 37 in Abstract, Line 166-167 in Results and Line 301-302 in Discussion. Moreover, a section was added in Discussion (Line 304-309) and the involvements of the concerned AMGs in other metabolic pathways were supplemented, as suggested.

On Line 304-309 in Discussion, we write:

*“However, several AMGs in the roTCA cycle, including *korB*, *frdA* and *ppdK*, could occur in the TCA cycle, as the roTCA cycle shares all enzymes with the TCA cycle^{37,38}. The AMGs *korB* and *frdA* were also involved in glutamate biosynthesis^{39,40} and fumarate respiration⁴¹, respectively. Therefore, it should be noted that these viral AMGs may contribute to C fixation and also other metabolic pathways.”*